# Client-Level Defense Placement for Adversarially Robust Federated Reinforcement Learning

**Anish Ambreth**                  *anish.ambreth@mbzuai.ac.ae*
*Department of Machine Learning*
*Mohamed Bin Zayed University of Aritficial Intelligence*

**Naveen Kumar Kummari**              *Naveen.Kummari@mbzuai.ac.ae*
*Department of Machine Learning*
*Mohamed Bin Zayed University of Aritficial Intelligence*

**Mohsen Guizani**                 *Mohsen.Guizani@mbzuai.ac.ae*
*Department of Machine Learning*
*Mohamed Bin Zayed University of Aritficial Intelligence*

**Reviewed on OpenReview:** *https://openreview.net/forum?id=JRpLJhvSCY*

## Abstract

Federated Reinforcement Learning (FRL) extends federated learning to sequential decision-making, enabling multiple clients to collaboratively train a global policy without sharing raw trajectories. While this setting is promising for privacy-sensitive domains such as autonomous systems and IoT control, it introduces critical attack surfaces: adversaries can corrupt policy gradients, and adaptive attackers that reshuffle targets and prioritize high-impact clients render static defenses brittle. Defenses in FRL operate at two complementary layers: server-side aggregation and client-level placement, but the latter remains under-formalized despite directly shaping attacker incentives. We propose FRL-CDPS (**C**lient-Level **D**efense **P**lacement for Adversarially Robust **F**ederated **R**einforcement **L**earning: A **S**tackelberg Approach), which models budget-constrained client-level defense placement as a Stackelberg game: the defender commits to a protection strategy while a rational Bayesian attacker best-responds under imperfect reconnaissance, maintaining posterior beliefs over each client's defense status. The framework captures partial observability and probabilistic defense effectiveness, faithfully reflecting real-world conditions where defenses are imperfect and adversaries operate under uncertainty. Despite NP-hardness of the defender's bilevel problem, we provide tractable solvers, namely exact feasible-set search for small systems and candidate-based Monte Carlo search for larger ones, with a $\frac{1}{2}$-approximation guarantee for the attacker oracle. Experiments on CartPole-v1, HalfCheetah-v2, and Walker2d-v5 across seven ablation dimensions show that FRL-CDPS consistently outperforms heuristic client-selection baselines (random, UCB, Thompson sampling) and composes effectively with server-side defenses (FLTG, FedGreed), demonstrating that Stackelberg planning provides a principled and practical advantage for client-level defense in FRL.

## 1 Introduction

Reinforcement learning (RL) has achieved remarkable results across domains such as gaming, robotics, and healthcare (Mnih et al., 2015; Silver et al., 2016; Li, 2017), yet real-world deployment is routinely constrained by sample efficiency: a single agent often lacks sufficient trajectories to learn a high-quality policy. Federated learning (FL) addresses this by enabling collaborative model training across many participants without exchanging raw data (McMahan et al., 2017; Konečnỳ et al., 2016; Sheller et al., 2020; Li et al., 2019). Federated Reinforcement Learning (FRL) lifts this paradigm to sequential decision-making (Qi et al., 2022), allowing geographically distributed agents, such as autonomous vehicles, industrial robots, or mobile devices, to jointly refine a global policy while keeping sensitive trajectory data local. This combination of sample

efficiency and privacy is essential in large-scale applications including autonomous driving, IoT control, personalization, and network resource management (Yang et al., 2019; Qi et al., 2022; Fang et al., 2025; Jiang et al., 2025).

Despite its promise, FRL exposes a critical and underexplored attack surface. Unlike supervised FL, where poisoning typically targets data labels or model weights (Bagdasaryan et al., 2020; Xie et al., 2020), FRL adversaries can manipulate *policy gradients* directly. Because gradients steer the global policy update at every round, even moderate corruptions can cascade through the aggregation step and severely destabilize the learned policy (Huang et al., 2017; 2021; Sun et al., 2020). Recent work demonstrates that these attacks are not merely theoretical: an adversary controlling a small fraction of clients can craft gradient perturbations that exploit the temporal structure of RL training, causing persistent and compounding policy degradation (Ma et al., 2023). Sophisticated multi-round attacks that enforce consistency across malicious clients can break many state-of-the-art server-side defenses simultaneously (Xie et al., 2025).

What makes FRL particularly challenging to defend is the adaptive nature of realistic adversaries. A static attacker who targets a fixed set of clients is, in practice, the easiest threat to handle. In real deployments, adversaries continuously re-optimize their strategy: reallocating attack budgets across rounds, periodically reassigning which clients to poison, and concentrating resources on high-value targets whose compromise inflicts the greatest harm to the global policy (Qi et al., 2022; Kumar et al., 2020; Han et al., 2021). This adaptive, temporal behavior exploits the sequential nature of FRL training and renders conventional defenses, which assume a fixed threat model, progressively less effective as training continues (Yang et al., 2019). At the same time, the defender must respond to evolving attacker behavior under a limited protection budget, giving rise to a cyclic strategic interaction that neither pure heuristics nor static optimization can faithfully capture.

Existing defenses in FRL and FL operate at two complementary but largely independent layers. *Server-side aggregation* defenses, such as coordinate-wise median (Yin et al., 2018), Krum (Blanchard et al., 2017), FLTG (Wen et al., 2025), and FedGreed (Kritharakis et al., 2025), filter or downweight suspicious gradient updates before aggregation. These methods have been refined extensively and offer meaningful robustness against certain attack patterns (Allouah et al., 2024; Fang et al., 2024). *Client-level placement* defenses determine which clients to enroll in each round, and have been approached using bandit heuristics such as UCB and Thompson sampling (Deressa & Hasan, 2024). However, these two layers have been studied in isolation. Server-side filters do not determine which clients to protect in the first place, and client-selection heuristics are purely reactive: they learn from past observations rather than anticipating how their selection will shape the attacker's incentives. Critically, no existing framework treats client-level defense placement as a *strategic commitment problem* under a budget, despite the fact that which clients are defended is the single most direct lever a defender has over the attacker's targeting decisions.

This gap motivates a game-theoretic formulation. The interaction between defender and attacker is inherently asymmetric and sequential: the defender, as leader, must commit to a client-level protection strategy before the attacker observes it and responds optimally. This structure is precisely the Stackelberg game model, which has been foundational in physical security domains (Conitzer & Sandholm, 2006; Tambe, 2011) and is now gaining traction in federated learning security (Li et al., 2024b). However, existing Stackelberg approaches in FL focus on reweighting aggregation or incentive design (Li et al., 2024b; Javaherian et al., 2025), not on the budget-constrained placement of client-level protections in FRL. Moreover, they do not model the FRL-specific structure: per-client defense and attack costs, gradient-noise injection attacks, partial reconnaissance of defense status by the attacker, and probabilistic defense effectiveness. Filling this gap requires a purpose-built framework that brings Stackelberg planning directly to the client-level placement problem in FRL.

To address this, we propose **FRL-CDPS** (**C**lient-Level **D**efense **P**lacement for Adversarially Robust **F**ederated **R**einforcement **L**earning: A **S**tackelberg Approach), a framework that models client-level defense allocation as a budget-constrained Stackelberg game. The defender, acting as leader, commits to a protection strategy: a deterministic set of clients to defend, subject to a total defense budget. The attacker, acting as follower, observes this commitment only through noisy reconnaissance and selects which clients to poison under its own budget, best-responding under imperfect information. The framework explicitly

accounts for partial observability, where the attacker receives only noisy signals of the defender's actions, and for probabilistic defense effectiveness, where a defended client can still be compromised with reduced probability. This captures the realistic setting where neither perfect monitoring nor guaranteed protection is available. We establish the theoretical foundations of the game, provide tractable solvers, and design FRL-CDPS as a modular layer that is orthogonal to and composable with existing server-side aggregation defenses.

Our contributions are as follows:

- To our knowledge, FRL-CDPS is the first framework that explicitly models client-level defense placement in FRL as a budget-constrained Stackelberg game, where the defender commits to a protection strategy and a rational Bayesian attacker best-responds under imperfect reconnaissance.

- We establish the theoretical foundations of the game: existence of Stackelberg equilibria (Lemma 1), the leader's commitment advantage over simultaneous-move Nash (Claim 1), reduction of the attacker's best response to a 0/1 knapsack (Lemma 2), NP-hardness of the defender's bilevel problem (Claim 3), and a $\frac{1}{2}$-approximation guarantee for the attacker oracle (Claim 2).

- We provide practical solvers, exact feasible-set search for small systems ($K \leq 15$) and a candidate-based Monte Carlo search for larger ones, designed as a modular layer orthogonal to and composable with existing server-side aggregation defenses.

- We evaluate FRL-CDPS on CartPole-v1, HalfCheetah-v2, and Walker2d-v5 across seven ablation dimensions (reshuffle frequency, attack style, budget regime, number of clients, attack intensity, observation accuracy, and attack regime), consistently outperforming client-selection heuristics (random, UCB, Thompson sampling) under adaptive multi-client attacks and composing effectively with server-side baselines (FLTG, FedGreed).

## 2 Related Work

**Server-side aggregation defenses.** A major line of FL security research studies robustness at the server aggregation layer. FLTG (Wen et al., 2025) is an angle-based Byzantine-robust aggregation method designed for non-IID federated learning: it combines a trusted or dynamically selected reference update, ReLU-clipped cosine-similarity filtering, norm alignment, and non-IID-aware weighting to suppress malicious or highly misaligned clients. FedGreed (Kritharakis et al., 2025) is a loss-based Byzantine-robust aggregation method: the server evaluates client updates on a trusted reference dataset, ranks clients by that loss, and greedily retains a low-loss subset for aggregation. Other representative methods include Krum (Blanchard et al., 2017) and coordinate-wise median (Yin et al., 2018), which filter statistical outliers among client updates. While effective against certain static attack patterns, these approaches do not optimize *which clients* to protect, leaving the client-level placement problem unaddressed. We use FLTG and FedGreed as server-side baselines since they represent complementary filtering philosophies (trust-score reweighting vs. greedy trusted-subset selection) and compose cleanly with our client-level defense layer.

**Client-level selection heuristics.** A parallel line of work addresses which clients to involve in each round. UCB-style selection policies (Khajehali et al., 2025; Waref et al., 2025) and Thompson-sampling variants (Deressa & Hasan, 2024) treat client participation as a bandit problem, maximizing utility based on past observations. These approaches are adaptive but reactive: they learn from observed outcomes rather than anticipating the attacker's response to the defender's commitment. They also lack budget constraints tied to per-client defense costs and carry no formal guarantee under adversarial non-stationarity. We adapt UCB and Thompson sampling to our setting under a shared budgeted defense interface, so that comparisons directly isolate the effect of Stackelberg planning versus bandit heuristics.

**FRL threat modeling and game-theoretic planning.** Recent work documents adversarial fragility in FRL pipelines (Huang et al., 2021), and game-theoretic approaches have been applied to FL incentive design (Jia et al., 2023; Li et al., 2024a; Hu et al., 2024; Zhang et al., 2022). Stackelberg commitment models are well-studied in physical security games (Conitzer & Sandholm, 2006; Tambe, 2011; von Stengel & Zamir,

2010), where a defender deploys limited resources against a best-responding adversary. However, these classical formulations assume a fixed, fully-specified defender resource set and do not model the FRL-specific structure: budget-constrained client-level protection, gradient-noise injection attacks, partial reconnaissance of defense status, and non-stationary client costs. FRL-CDPS instantiates Stackelberg planning directly in this setting with explicit attacker best-response oracles and formal complexity results.

**Positioning.** Table 1 summarizes our scope: server aggregation and client-level placement are complementary control layers, and FRL-CDPS contributes a strategic planning formulation for the client-level layer.

Table 1: Scope comparison across defense families. *Client-Set Opt.*: explicitly selects which clients to protect each round. *Joint Atk+Defense*: formulates attacker and defender as coupled strategic agents. FRL-CDPS is the only method satisfying both criteria.

| Family | Representative Methods | Client-Set Opt. | Joint Atk+ Defense |
|---|---|---|---|
| Server aggregation | FLTG (Wen et al., 2025), FedGreed (Kritharakis et al., 2025) | No | No |
| Client heuristics | Random, UCB (Khajehali et al., 2025; Waref et al., 2025), Thompson Sampling (Deressa & Hasan, 2024) | Yes | No |
| FRL-CDPS (ours) | Stackelberg game-based client-level placement | Yes | Yes |

**Theoretical novelty relative to prior work.** FRL-CDPS does not propose a new robust aggregation rule or a new reactive client-selection heuristic. Its theoretical contribution is to formalize client-level defense placement itself as a finite Stackelberg security game under budget constraints, noisy reconnaissance, and probabilistic defense effectiveness. This formulation yields a distinct set of results: existence of deterministic Stackelberg equilibria, reduction of the attacker's best response to a 0/1 knapsack problem, NP-hardness of the defender's bilevel optimization problem, and a $\frac{1}{2}$-approximation guarantee for the attacker oracle. Prior FL and FRL defenses primarily study filtering, reweighting, or reactive selection; our analysis characterizes the commitment structure and computational properties of budgeted client-level protection against an adaptive follower.

## 3 Background

We study a synchronous cross-client federated reinforcement learning (FRL) setup where multiple clients collaborate to train a shared global policy without exchanging raw trajectories (McMahan et al., 2017; Konečný et al., 2016; Yang et al., 2019). Here we focus on the technical setup underlying our framework.

### 3.1 Federated Reinforcement Learning Setup

We consider a synchronous cross-client federated reinforcement learning (FRL) setup with $K$ clients $\mathcal{C} = \{1, \ldots, K\}$. At federated round $t$, the server broadcasts the current global policy $\pi_{\theta^{(t-1)}}$ to all clients. Each client $i$ then interacts with an independent local instance of a common Markov decision process (MDP), with shared state space $\mathcal{S}$, action space $\mathcal{A}$, and reward objective. In the clean baseline, these local rollouts are identically distributed across clients conditional on $\pi_{\theta^{(t-1)}}$, since all clients interact with independent copies of the same task. We do not model client heterogeneity through different underlying MDPs; instead, heterogeneity enters through the strategic parameters $\{w_i, c_{d,i}, c_{a,i}\}$ and through which clients are attacked or defended.

A single trajectory collected by client $i$ at round $t$ is denoted by

$$\tau_{i,n}^{(t)} = \big(s_{i,n,0}^{(t)}, a_{i,n,0}^{(t)}, r_{i,n,0}^{(t)}, \ldots, s_{i,n,H}^{(t)}\big), \tag{1}$$

where $n$ indexes trajectories collected by client $i$ within round $t$, $\ell$ indexes within-trajectory time steps, and $H$ is the trajectory horizon. The full local batch collected by client $i$ at round $t$ is

$$\mathcal{D}_i^{(t)} = \{\tau_{i,n}^{(t)}\}_{n=1}^{N_i^{(t)}}. \tag{2}$$

For compactness, when defining the FRL objective we use $\tau$ to denote a generic trajectory drawn under policy $\pi_\theta$:

$$J(\pi_\theta) = \mathbb{E}_{\tau \sim \pi_\theta} \left[ \sum_{\ell=0}^{H-1} r_\ell \right], \tag{3}$$

Thus, $\tau$ denotes one generic trajectory, while $\mathcal{D}_i^{(t)}$ denotes the batch of trajectories used by client $i$ at round $t$. From $\mathcal{D}_i^{(t)}$, each client computes a policy-gradient estimate $g_i^{(t)}$. For discrete-action environments (CartPole-v1) we use REINFORCE (Williams, 1992), and for continuous-action environments (HalfCheetah-v2 and Walker2d-v5) we use a PPO-style clipped surrogate objective (Schulman et al., 2017) with clip parameter $\epsilon = 0.2$.

A central server then aggregates local gradients using FedAvg (McMahan et al., 2017) and applies the global update

$$\theta^{(t)} \leftarrow \theta^{(t-1)} + \gamma \cdot \frac{1}{K} \sum_{i=1}^{K} g_i^{(t)}, \tag{4}$$

where $\gamma$ is the global learning rate. This aggregation rule is fixed throughout our framework, and our optimization variable is exclusively the client-level protection set. Adversarial dynamics and the client-level defense-selection layer are introduced on top of this synchronous FRL baseline in Section 4.

We model adversarial interactions as a Stackelberg security game (Tambe, 2011; Conitzer & Sandholm, 2006): the defender (leader) commits to a budget-constrained client-protection strategy first, and the attacker (follower) best-responds given noisy reconnaissance of that commitment. Formal game definitions, utility functions, and equilibrium characterization are given in Section 4.

### 3.2 Threat Model

We consider a setting where, in each synchronous federated round, an external attacker may compromise any subset of clients whose total attack cost is at most $B_A$ (Huang et al., 2021; Zhang et al., 2020; Qi et al., 2022). Thus, the number of adversarial clients is not fixed by a Byzantine fraction in advance; instead, it is determined endogenously by the attacker's budget and per-client costs, yielding the single-client and multi-client regimes formalized below. Compromise status is not permanent across rounds. Because the attacker recomputes its best response every round, a client that is benign in one round may be compromised in a later round if selected by the attacker, while a previously attacked client reverts to benign behavior in rounds in which it is not selected. The server does not observe ground-truth benign/adversarial labels. It only receives client updates and may apply probabilistic validation or robust aggregation rules, so detection is imperfect rather than oracle-based. We also do not model endogenous "role contagion," where an honest client becomes intrinsically malicious solely because the shared global policy has been degraded; such clients remain benign in our formulation, even if degraded policies cause them to generate lower-quality trajectories. Following Section 3.1, each client $C_i$ holds a local trajectory batch $\mathcal{D}_i^{(t)}$ at round $t$, and each trajectory $\tau_{i,n}^{(t)} \in \mathcal{D}_i^{(t)}$ is a sequence of transitions $(s_{i,n,\ell}^{(t)}, a_{i,n,\ell}^{(t)}, r_{i,n,\ell}^{(t)}, s_{i,n,\ell+1}^{(t)})$. All three attack types share a common high-level structure: adversarial client $C_i$, when successfully compromised, corrupts the training signal before it contributes to the global update. The attacks differ in *where* the corruption is injected. For readability, the attack equations below suppress the round index $t$ and trajectory index $n$ when they are not essential.

**(i) Gradient noise injection** (post-computation; Bhagoji et al. 2019): The adversarial client perturbs computed policy gradients before aggregation:

$$\nabla_i' = \nabla_i + \eta \frac{w_i}{w_{\text{ref}}} \big( \max(\nabla_i) - \min(\nabla_i) \big) \boldsymbol{\xi}_i \tag{5}$$

where $\nabla_i$ is the true gradient, $\eta \geq 0$ is the attack-intensity multiplier, $\boldsymbol{\xi}_i$ has i.i.d. Uniform$(-1, 1)$ entries, $(\max(\nabla_i) - \min(\nabla_i))$ is the gradient range that adaptively scales noise to each parameter's natural magnitude, $w_i > 0$ is the damage weight, and $w_{\mathrm{ref}} > 0$ is a normalization reference. High-$w_i$ clients inject proportionally more disruptive noise.

**(ii) Action flip** (at trajectory collection, pre-computation; Huang et al. 2017; Lin et al. 2017): The adversarial client substitutes locally-sampled actions with adversarial alternatives during environment interaction:

$$a'_\ell = \arg\min_{a \in \mathcal{A}} \pi_\theta(a \mid s_\ell), \tag{6}$$

selecting the action least favored by the current policy at each step. This generates a corrupted trajectory $\mathcal{D}'_i$ whose policy gradient points away from the current policy, without requiring any access to $\theta$ beyond the action distribution at the current state.

**(iii) Reward poisoning** (at trajectory collection, pre-computation; Ma et al. 2019): The adversarial client negates and scales the observed rewards before local gradient computation:

$$r'_\ell = -\eta \cdot \frac{w_i}{w_{\mathrm{ref}}} \cdot |r_\ell|, \tag{7}$$

causing the policy gradient to push the policy toward low-reward behaviors. The $w_i/w_{\mathrm{ref}}$ scaling preserves damage-weight proportionality across attack types.

All three attacks are conditionally applied only when the attack succeeds under the defense model formalized later in Eq. (11), preserving the unified game-theoretic formulation across attack types. These three attack families should be viewed as representative instantiations rather than an exhaustive taxonomy of FRL attacks: the same Stackelberg client-allocation layer can accommodate other client-side corruption models so long as they induce a per-client attack value and realized residual-damage effect. Throughout, $\eta \geq 0$ is the attack-intensity multiplier, $w_i > 0$ is the damage weight for client $C_i$, and $w_{\mathrm{ref}} > 0$ is a normalization reference.

**Damage Weights.** Federated systems naturally exhibit heterogeneity in client strategic importance: major hospital systems contribute more valuable data than individual clinics, gateway nodes are more critical than edge sensors, and large financial institutions have greater systemic impact than smaller participants (Li et al., 2020; Wang et al., 2021; Yang et al., 2019). Yet existing FL security analyses typically assume uniform client importance (Blanchard et al., 2017; Yin et al., 2018). We model this heterogeneity via damage weights $w_i > 0$: clients with larger $w_i$ amplify attack impact through the $w_i/w_{\mathrm{ref}}$ scaling shared across all three attack types, and the defender must prioritize protecting high-$w_i$ clients under budget constraints. The damage weight $w_i$ represents the strategic importance of client $i$. In practice, $w_i$ may be estimated from factors such as data volume, policy contribution, or historical training impact; in our framework it is treated as an exogenous input supplied to the defender.

**Adversarial objectives and capabilities.** The attacker aims to degrade global policy performance by maximizing the reduction in $J(\pi_\theta)$. It is *black-box with respect to model internals* (no access to $\theta$, local MDPs $\mathcal{M}_i$, or server aggregation logic), and its capability is limited to corrupting the training signal at compromised clients via gradient noise injection, action substitution, or reward manipulation (formalized above). Separately, the attacker *does* possess reconnaissance capability: it receives a noisy binary signal $o_i$ per client indicating whether that client appears to be defended ($q \in (0, 1]$ is the accuracy of this signal). The attacker acts as a rational Bayesian agent over this defense-placement signal, maintaining posterior beliefs over each client's defense status and best-responding accordingly. These two aspects are orthogonal: black-box refers to the training process, while Bayesian reasoning applies only to the strategic defense-placement game.

**Attack strategy.** At each federated round, the attacker re-solves its budget-constrained best-response problem and selects a subset $s_A \subseteq \mathcal{C}$ of clients to compromise. Thus, the attack set is dynamic rather than static across training. The selection prioritizes clients with higher expected impact under current costs, defense decisions, and observation uncertainty. The attacker may periodically reshuffle compromised clients over rounds to maintain degradation and avoid persistent targeting patterns.

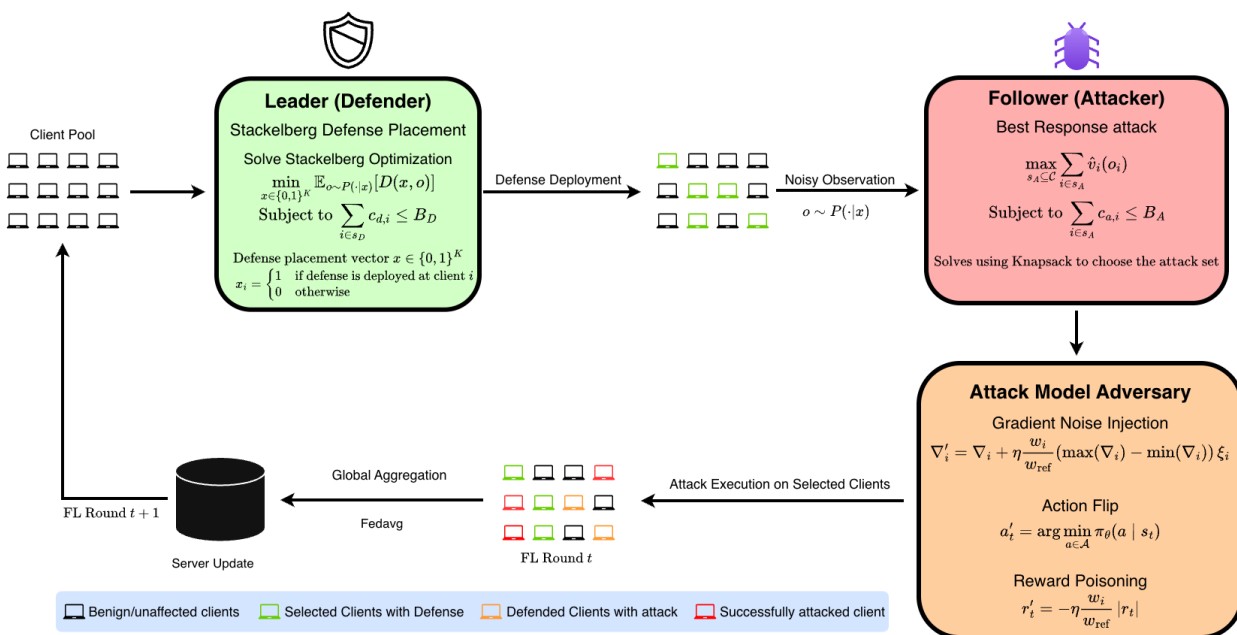

Figure 1: The FRL-CDPS training pipeline illustrating the budget-constrained Stackelberg game at the client-selection layer. In each federated round, the server-side defender (leader) commits to a protection strategy vector $x \in \{0,1\}^K$ subject to a defense budget $B_D$. The attacker (follower) receives a noisy reconnaissance signal $o \sim P(\cdot|x)$ indicating which clients appear defended. Acting as a rational Bayesian agent, the attacker computes the estimated value of compromising each client and solves a 0/1 knapsack problem to select the optimal attack set under budget $B_A$. During local training and global aggregation via FedAvg, deployed defenses probabilistically reduce the success rate of the chosen attacks. The server then updates the global policy and advances to the next training round.

**Definition 1** (Adversarial Regimes). *We study four adversarial regimes that vary along two axes:*

- ***Cardinality:*** Single-client *($|s_A| = 1$ at each round) vs.* multi-client *($|s_A| > 1$, bounded by attacker budget $B_A$).*

- ***Persistence:*** Static *(game parameters $\{w_i, c_{d,i}, c_{a,i}\}$ are fixed for all rounds) vs.* reshuffling *(damage weights $w_i$, defense costs $c_{d,i}$, and attack costs $c_{a,i}$ are jointly resampled every $T_{\text{reshuffle}}$ rounds, forcing bandit-style defenses to re-learn the client landscape from scratch).*

*Crossing these two axes yields four regimes: (i) single-client static, (ii) single-client reshuffling, (iii) multi-client static, and (iv) multi-client reshuffling. Reshuffling models realistic non-stationarity in federated systems: client hardware profiles, network conditions, and data distributions shift over time, altering both the cost of defending each client and the strategic value of compromising it. FRL-CDPS handles this gracefully by recomputing its Stackelberg allocation from updated parameters, whereas bandit methods must restart exploration after each reshuffle. Equivalently, in the static regime the tuples $(w_i, c_{d,i}, c_{a,i})$ remain fixed throughout training, while in the reshuffling regime they are piecewise constant within an era and jointly resampled every $T_{\text{reshuffle}}$ rounds.*

## 4 Framework of FRL-CDPS

FRL-CDPS models client-level defense as a two-player Stackelberg game between a server-side defender (leader) and a budget-constrained attacker (follower), as illustrated in Figure 1. We now formalize each component.

Table 2: Notation summary for FRL-CDPS.

| Symbol | Meaning |
| --- | --- |
| $K$ | Number of clients |
| $\mathcal{C} = \{1, \ldots, K\}$ | Set of clients |
| $s_D, s_A$ | Subsets of clients chosen for defense / attack |
| $c_{d,i}, c_{a,i}$ | Defense cost / attack cost for client $i$ |
| $B_D, B_A$ | Defender / attacker budget |
| $w_i$ | Damage weight (strategic importance of client $i$) |
| $w_{\mathrm{ref}}$ | Reference weight for normalizing damage scaling |
| $U_D, U_A$ | Defender and attacker utility functions |
| $x_i, y_i$ | Binary defense / attack indicators for client $i$ |
| $\mathcal{H}(x)$ | Defender's expected residual damage; $\mathcal{H}(x) = \mathbb{E}_o[D(x, o)]$ |
| $D(x, o)$ | Realized damage under defense $x$ and attacker observation $o$ |
| $\tau_{i,n}^{(t)}$ | One trajectory collected by client $i$ at round $t$ |
| $\mathcal{D}_i^{(t)}$ | Local batch of trajectories collected by client $i$ at round $t$ |
| $N_i^{(t)}$ | Number of trajectories collected by client $i$ at round $t$ |
| $g_i^{(t)}$ | Policy-gradient estimate computed by client $i$ at round $t$ |
| $t, T$ | Round index / total number of training rounds |
| $\ell, n$ | Within-trajectory step index / local trajectory index in $\mathcal{D}_i^{(t)}$ |
| $T_{\mathrm{reshuffle}}$ | Reshuffle frequency (rounds between parameter resampling) |
| $H$ | Trajectory horizon |
| $\delta$ | Defense strength (probability that defense blocks an attack) |
| $q$ | Observation accuracy of the attacker's reconnaissance |
| $o_i$ | Attacker's noisy observation of client $i$'s defense status |
| $\pi_0$ | Attacker's prior belief that a client is defended; $\pi_0 = \min(1, B_D / \sum_j c_{d,j})$ |
| $\hat{p}_i(o_i)$ | Attacker's estimated attack-success probability for client $i$ |
| $\hat{v}_i(o_i)$ | Attacker's estimated value of compromising client $i$; $w_i \hat{p}_i(o_i)$ |
| $\eta$ | Attack-intensity multiplier |
| $\gamma$ | Learning rate for global policy update |
| $\pi_\theta$ | Global policy parameterized by $\theta$ |
| $J(\pi_\theta)$ | Expected return of policy $\pi_\theta$ |

## 4.1 Notation Summary

Before defining the Stackelberg game formally, Table 2 summarizes the main symbols used throughout the framework and experiments. Several entries are introduced in detail in the subsections that follow.

**Estimating deployment parameters.** In real deployments, the damage weights $w_i$ and cost parameters $c_{d,i}$ and $c_{a,i}$ can be estimated from observable system signals rather than taken as perfectly specified. The damage weight $w_i$ can be derived from client importance indicators such as data volume, historical contribution to validation return, update influence, task criticality, or the performance drop observed when client $i$ is excluded from training. The defense cost $c_{d,i}$ can be estimated from the marginal cost of protecting client $i$, including secure execution overhead, additional validation, communication latency, privacy-budget consumption, or manual audit cost. The attack cost $c_{a,i}$ can be specified as the defender's estimate of the adversary's relative difficulty or resource cost for compromising client $i$, using signals such as device trust level, network exposure, patch status, authentication strength, historical compromise alerts, or red-team assessments. These estimates can be refreshed at each reshuffle era, and uncertainty in them can be handled conservatively through sensitivity analysis or interval-valued costs.

## 4.2 Stackelberg FRL Game

We define $\mathcal{G}_{\mathrm{FRL}} = \langle \{D, A\}, S_D, S_A, U_D, U_A \rangle$, where $D$ is the defender (server-side security planner) and $A$ the attacker (malicious entity). The defender's action is client-level protection selection, while server aggregation is fixed. Each client $i$ has a defense cost $c_{d,i} \geq 0$ and an attack cost $c_{a,i} \geq 0$. The defender chooses a protection set $s_D \subseteq \mathcal{C}$ with budget

$$\sum_{i \in s_D} c_{d,i} \leq B_D. \tag{8}$$

The attacker chooses an attack set $s_A \subseteq \mathcal{C}$ with budget

$$\sum_{i \in s_A} c_{a,i} \leq B_A. \tag{9}$$

Let $o = (o_1, \ldots, o_K) \in \{0, 1\}^K$ denote the attacker's noisy observation of the defender's actions, where $o_i = 1$ indicates that client $i$ appears defended.

**Observation model.** Given the defender's binary allocation $x \in \{0, 1\}^K$ (where $x_i = 1$ iff $i \in s_D$), observations are generated independently per client via a symmetric binary channel:

$$P(o \mid x) = \prod_{i=1}^{K} P(o_i \mid x_i), \qquad P(o_i = x_i \mid x_i) = q, \quad P(o_i = 1 - x_i \mid x_i) = 1 - q, \tag{10}$$

where $q \in (0, 1]$ is the *observation accuracy* parameter. The channel flips the true defense status with probability $1 - q$ in *both* directions: a defended client ($x_i = 1$) falsely appears undefended with probability $1 - q$, and an undefended client ($x_i = 0$) falsely appears defended with probability $1 - q$.

**Attack-success probability.** Let $\delta \in [0, 1]$ be the *defense strength*. Defending client $i$ activates a validation mechanism (e.g., gradient anomaly detection) that causes the attack to fail with probability $\delta$, leaving an undefended client always compromised. Formally, the attack-success probability given true defense status $x_i$ is:

$$p_i(x_i) = 1 - \delta \cdot x_i. \tag{11}$$

The observation $o_i$ does not change the realized outcome, and enters only through the attacker's selection of which clients to attack.

**Attacker's Bayesian estimated value.** Since the attacker cannot observe $x_i$ directly, it maintains a prior $\pi_0 = \min(1, B_D / \sum_j c_{d,j}) \in (0, 1]$ (the fraction of total defense cost the budget can cover, clipped to 1 when the defender can afford all clients) as a uniform belief that any given client is defended. This prior is an approximation: it ignores heterogeneity in individual defense costs $c_{d,i}$ and treats all clients as equally likely to be protected. Formally, the attacker assumes $x_1, \ldots, x_K$ are *i.i.d.* Bernoulli($\pi_0$), independently across clients. It then applies Bayes' rule given observation $o_i$, treating each client's defense indicator as conditionally independent given $\pi_0$:

$$P(x_i = 1 \mid o_i = 1) = \frac{q \pi_0}{q \pi_0 + (1-q)(1-\pi_0)}, \tag{12}$$

$$P(x_i = 1 \mid o_i = 0) = \frac{(1-q)\pi_0}{(1-q)\pi_0 + q(1-\pi_0)}. \tag{13}$$

The attacker's estimated attack-success probability is:

$$\hat{p}_i(o_i) = 1 - \delta \cdot P(x_i = 1 \mid o_i), \tag{14}$$

$$\hat{v}_i(o_i) = w_i \, \hat{p}_i(o_i). \tag{15}$$

The realized utilities depend on true $x_i$, not on $o$. For a given attack set $s_A$ (selected via the estimated values above), the attacker's realized damage and the defender's residual damage are:

$$U_A(s_D, s_A) = \sum_{i \in s_A} w_i \, (1 - \delta \cdot x_i), \qquad U_D(s_D, s_A) = - \sum_{i \in s_A} w_i \, (1 - \delta \cdot x_i). \tag{16}$$

The observation $o$ does not appear in these expressions directly, and enters only through the attacker's selection of $s_A$ via $\hat{v}_i(o_i)$. Since $o$ is random, the defender optimizes the *expected* residual damage over observation realizations:

$$\bar{U}_D(s_D) = \mathbb{E}_{o \sim P(\cdot|x)}\Big[U_D\big(s_D, s_A^\star(o)\big)\Big], \tag{17}$$

where $s_A^\star(o) = \arg\max_{s_A} \hat{U}_A(s_A; o)$ is the attacker's best response to observation $o$, and $\hat{U}_A(s_A; o) = \sum_{i \in s_A} \hat{v}_i(o_i)$ is the attacker's estimated utility used for selection. Since $U_D(s_D, s_A) = -\sum_{i \in s_A} w_i(1 - \delta x_i)$ is the negative of realized damage, we have $\bar{U}_D(s_D) = -\mathcal{H}(x)$, where $\mathcal{H}(x) = \mathbb{E}_o[D(x, o)]$ denotes the expected realized damage (formalized as the defender's objective in Section 4.4). Maximizing $\bar{U}_D$ is therefore equivalent to minimizing $\mathcal{H}(x)$.

**Connection to $J(\pi_\theta)$.** The damage weight $w_i$ serves as a proxy for client $i$'s marginal contribution to the global return $J(\pi_\theta)$: corrupting a client with large $w_i$ injects proportionally more disruptive gradient noise (via the $w_i/w_{\text{ref}}$ scaling in the attack model), causing a larger reduction in $J(\pi_\theta)$. The game utilities $U_A$ and $U_D$ thus summarize the attacker's and defender's objectives in terms of expected policy damage, without requiring either player to know $\theta$ or $J$ directly.

**Assumptions:** (i) Probabilistic defense: defended clients can still be compromised with reduced success probability. (ii) Partial observability: the attacker observes a noisy signal of defended clients rather than exact defense actions. (iii) Rational players: both optimize expected utilities under their budget constraints.

**Definition 2** (Stackelberg Equilibrium). *A pair $(s_D^\star, s_A^\star)$ is a Stackelberg equilibrium if:*

1. ***Attacker best-responds*** *to each realized observation $o$:*

$$s_A^\star(o) \in R_A(s_D^\star, o) = \arg\max_{s_A: \sum_{i \in s_A} c_{a,i} \leq B_A} \hat{U}_A(s_A; o), \tag{18}$$

   *where $\hat{U}_A(s_A; o) = \sum_{i \in s_A} \hat{v}_i(o_i)$ uses the attacker's estimated values.*

2. ***Defender commits optimally*** *under budget, anticipating the distribution of attacker responses:*

$$s_D^\star \in \arg\max_{s_D: \sum_{i \in s_D} c_{d,i} \leq B_D} \mathbb{E}_{o \sim P(\cdot|s_D)}\Big[U_D\big(s_D, s_A^\star(o)\big)\Big], \tag{19}$$

   *where $P(\cdot \mid s_D)$ denotes the observation distribution induced by $s_D$ (i.e., $x_i = \mathbf{1}[i \in s_D]$ in the symmetric channel). Since $U_D = -D$ is the negative of damage, maximizing $\bar{U}_D$ is equivalent to minimizing expected damage $\mathcal{H}(x)$ as in the bilevel formulation in Eq. (21).*

**Lemma 1** (Equilibrium Existence). *At least one Stackelberg equilibrium in deterministic behavioral strategies exists for the finite strategy spaces above. (The defender plays a pure protection set $s_D^\star \subseteq \mathcal{C}$, and the attacker plays a deterministic map $s_A^\star : \{0,1\}^K \to 2^{\mathcal{C}}$ from observations to attack sets.)*

*Proof.* We establish existence by showing strategy spaces are finite, utilities are well-defined, and both optimization problems attain their optima.

**Finiteness.** The defender's budget-feasible strategy space $S_D \subseteq 2^{\mathcal{C}}$ satisfies $|S_D| \leq 2^K < \infty$. Similarly $|S_A| \leq 2^K < \infty$.

**Well-defined utilities.** $U_A(s_D, s_A)$ and $U_D(s_D, s_A)$ are real-valued and bounded on $S_D \times S_A$ since all $w_i$ and $\delta$ are finite. For any fixed $o$, $\hat{U}_A(s_A; o) = \sum_{i \in s_A} \hat{v}_i(o_i)$ is also bounded.

**Attacker's best response.** For any $s_D$ and realized $o$, maximizing $\hat{U}_A$ over the finite set $S_A$ attains a maximum, so $R_A(s_D, o) \neq \emptyset$.

**Defender's optimization.** The defender maximizes $\bar{U}_D(s_D) = \mathbb{E}_o[U_D(s_D, s_A^\star(o))]$ over the finite set $S_D$. Since $S_D$ is finite and $\bar{U}_D$ is well-defined and bounded, a maximizer $s_D^\star$ exists.

**Equilibrium construction.** With $s_D^\star$ fixed, select $s_A^\star(o) \in R_A(s_D^\star, o)$ for each $o$. The pair $(s_D^\star, s_A^\star(\cdot))$ satisfies both conditions of Definition 2 and constitutes a Stackelberg equilibrium in deterministic behavioral strategies. $\square$

**Claim 1** (Leader's Advantage). *Under standard Stackelberg security-game assumptions with defender-favorable equilibrium selection, the defender's commitment value weakly dominates the corresponding simultaneous-move benchmark.*

*Proof.* In the simultaneous-move (Nash) game, suppose the equilibrium strategies are $(s_D^N, s_A^N)$. In the Stackelberg game, the defender can always commit to $s_D^N$, which induces the same follower best-response set. With defender-favorable tie-breaking, the resulting utility is at least $U_D(s_D^N, s_A^N)$. Since the defender can additionally optimize over all commitment strategies, the Stackelberg value is weakly higher: $\bar{U}_D(s_D^\star) \geq U_D(s_D^N, s_A^N)$. This is the classical commitment-effect result in security games (Conitzer & Sandholm, 2006; Tambe, 2011; von Stengel & Zamir, 2010). In our optimization we use pessimistic tie-breaking (worst-case over follower best responses), which provides a lower bound on the commitment value. $\square$

### 4.3   Attacker's Problem: Knapsack Reduction

We establish that the attacker's optimization reduces to a standard 0/1 knapsack problem.

**Lemma 2** (Knapsack Reduction). *For fixed defender strategy $s_D$ and observation $o$, the attacker's best-response problem*

$$\max_{s_A \subseteq \mathcal{C}} \quad \sum_{i \in s_A} \hat{v}_i(o_i)$$

$$s.t. \quad \sum_{i \in s_A} c_{a,i} \leq B_A$$

*is a 0/1 knapsack with values $\hat{v}_i(o_i) = w_i \hat{p}_i(o_i) \geq 0$, weights $c_{a,i}$, and capacity $B_A$.*

*Proof.* The attacker's objective $\hat{U}_A(s_A; o) = \sum_{i \in s_A} \hat{v}_i(o_i)$ is a non-negative linear sum over a binary selection variable $y_i \in \{0, 1\}$ with $y_i = \mathbf{1}[i \in s_A]$, subject to a single linear budget constraint. For any client with non-positive estimated value, including it in $s_A$ provides no gain while consuming budget, so we restrict to $I^+ = \{i \in \mathcal{C} : \hat{v}_i(o_i) > 0\}$ without loss.

Define the bijection $\phi : 2^{I^+} \leftrightarrow \{0, 1\}^{|I^+|}$ (using indicator variable $z$ to avoid collision with the defense allocation vector $x$):

$$\phi(s_A) = z \text{ where } z_i = \begin{cases} 1 & \text{if } i \in s_A \\ 0 & \text{otherwise} \end{cases}$$

For any $s_A \subseteq I^+$ and $z = \phi(s_A)$:

$$\sum_{i \in s_A} \hat{v}_i(o_i) = \sum_{i \in I^+} \hat{v}_i(o_i) \, z_i, \qquad \sum_{i \in s_A} c_{a,i} \leq B_A \iff \sum_{i \in I^+} c_{a,i} z_i \leq B_A.$$

Since $\phi$ is bijective and preserves both objective values and feasibility, the attacker's problem is precisely a 0/1 knapsack with item values $\hat{v}_i(o_i)$, weights $c_{a,i}$, and capacity $B_A$. $\square$

**Claim 2** (Computational Complexity). *Assuming attack costs $c_{a,i}$ and budget $B_A$ are rational (scaled to integers), the attacker's problem can be solved exactly in $O(|I^+| \cdot B_A)$ pseudo-polynomial time via dynamic programming, where $I^+ = \{i : \hat{v}_i > 0\}$, or approximated with a $\frac{1}{2}$-factor guarantee in $O(|I^+| \log |I^+|)$ time using greedy+best-single selection (assuming $c_{a,i} > 0$ for all $i$, so that item densities $\hat{v}_i/c_{a,i}$ are well-defined).*

*Proof.* Standard knapsack DP uses table $\text{DP}[i, b] = $ max utility from first $i$ items with budget $b$, and items with $\hat{v}_i \leq 0$ are excluded (set $I^+$) without loss. For approximation, we use the *greedy+best-single-item* rule: compare (i) density-ordered greedy packing and (ii) the best feasible singleton.

$\frac{1}{2}$**-approximation guarantee.** Let $v_{\max} := \max_{i \in I^+} \hat{v}_i$ and $v_{\text{frac}}$ denote the optimal fractional knapsack value. Ordering items by density $\rho_i = \hat{v}_i/c_{a,i}$, let $V_G$ be the total value of the maximal prefix that fits under capacity. If item $j$ is the first item that does not fit completely, then the fractional solution that takes a

fraction $\lambda \in (0,1)$ of item $j$ achieves $V_G + \lambda \hat{v}_j = v_{\text{frac}}$. Thus $v_{\text{frac}} \leq V_G + v_{\text{max}}$, so $V_G \geq v_{\text{frac}} - v_{\text{max}}$. The augmented greedy oracle returns

$$v_{\text{greedy}} = \max\{V_G, v_{\text{max}}\} \geq \max\{v_{\text{max}}, v_{\text{frac}} - v_{\text{max}}\}. \tag{$\star$}$$

Let $v_{\text{int}}$ be the optimal integral value. (i) If $v_{\text{int}} \leq 2v_{\text{max}}$, then $v_{\text{max}} \geq \frac{1}{2}v_{\text{int}}$, and by ($\star$) we get $v_{\text{greedy}} \geq \frac{1}{2}v_{\text{int}}$. (ii) If $v_{\text{int}} > 2v_{\text{max}}$, then $v_{\text{frac}} \geq v_{\text{int}}$ and $v_{\text{frac}} - v_{\text{max}} > \frac{1}{2}v_{\text{int}}$, so by ($\star$), $v_{\text{greedy}} \geq \frac{1}{2}v_{\text{int}}$. This follows the classical analysis of Martello & Toth (1990). $\square$

## 4.4 Defender's Bilevel Optimization

We encode the defender's protection strategy with a binary vector $x \in \{0,1\}^K$, where $x_i = 1$ means client $i$ is protected. The attacker observes a noisy signal $o \sim P(\cdot|x)$ and selects a binary attack vector $y \in \{0,1\}^K$ using its Bayesian estimated values $\hat{v}_i(o_i) = w_i \hat{p}_i(o_i)$.

**Inner problem (attacker best response for realized $o$).** Given $x$ and observation $o$, the attacker solves:

$$y^\star(o) = \arg \max_{y \in \{0,1\}^K} \sum_{i=1}^K \hat{v}_i(o_i)\, y_i \tag{20}$$

$$\text{s.t.} \quad \sum_{i=1}^K c_{a,i}\, y_i \leq B_A.$$

This is a 0/1 knapsack on items $i \in \mathcal{C}$ with values $\hat{v}_i(o_i)$, weights $c_{a,i}$, and capacity $B_A$ (Lemma 2). The *realized damage* from attacker response $y^\star(o)$ is $D(x,o) = \sum_i w_i(1 - \delta x_i)y_i^\star(o)$.

**Outer problem (defender planning).** The defender minimizes expected residual damage over the distribution of observations:

$$\min_{x \in \{0,1\}^K} \quad f(x) = \mathcal{H}(x) = \mathbb{E}_{o \sim P(\cdot|x)}\big[D(x,o)\big] \tag{21}$$

$$\text{s.t.} \quad \sum_{i=1}^K c_{d,i}\, x_i \leq B_D.$$

Eqs. (20)–(21) precisely capture the Stackelberg structure: the defender (leader) commits to $x$, the attacker (follower) responds to each observation draw, and the defender anticipates the distribution of attacker responses via the expectation in $\mathcal{H}(x)$.

**Claim 3** (NP-hardness of Defender's Problem)**.** *The defender's bilevel optimization problem in Eq. (21) is NP-hard.*

*Proof.* We reduce from the 0/1 knapsack decision problem (KDP). An instance is $(\alpha_i, v_i, W, V)$ with item costs $\alpha_i > 0$, values $v_i \geq 0$, capacity $W$, and target $V$: does there exist $z \in \{0,1\}^K$ with $\sum_i \alpha_i z_i \leq W$ and $\sum_i v_i z_i \geq V$?

*Construction.* For each item $i$, create a client with $c_{d,i} = \alpha_i$, $c_{a,i} = 0$, $w_i = v_i$, and set $B_D = W$, $B_A = 0$. Set $\delta = 1$ and $q = 1$ (perfect defense and perfect observation). Then $o_i = x_i$ exactly, and the Bayesian posterior gives $P(x_i = 1 \mid o_i) = x_i$, so $\hat{v}_i(o_i) = w_i(1 - x_i)$. Since $c_{a,i} = 0$, the attacker can always attack all undefended clients, so $\mathcal{H}(x) = \sum_i w_i(1 - x_i) = \sum_i v_i - \sum_i v_i x_i$.

*Equivalence.* Since $\sum_i v_i$ is constant, minimizing $\mathcal{H}(x)$ subject to $\sum_i \alpha_i x_i \leq W$ is equivalent to maximizing $\sum_i v_i x_i$ under the same budget, which is exactly the 0/1 knapsack optimization problem. For the decision version, define threshold $L = \sum_i v_i - V$, so that $\mathcal{H}(x) \leq L \iff \sum_i v_i x_i \geq V$ under $\sum_i \alpha_i x_i \leq W$, so solving the defender's decision problem solves KDP. The reduction is polynomial-time, so the defender's problem is NP-hard. $\square$

*Approximation oracle.* The attacker oracle achieves a $\frac{1}{2}$-approximation per observation sample $o$ for the attacker's *estimated* knapsack objective (Claim 2). That is, the fallback oracle returns an attack set whose estimated utility $\hat{U}_A(s_A; o)$ is at least one half of the optimal estimated best-response value for that observation. The realized damage $D(x, o)$ is then computed using the true defense status $x$, so under noisy reconnaissance this guarantee should be interpreted as an attacker-oracle guarantee rather than as a global approximation bound on the defender's residual damage $\mathcal{H}(x)$. In the outer search, each candidate defense set is still ranked by Monte Carlo estimates of realized residual damage using the attack set returned by the oracle.

**Algorithmic Solutions.** Since exact defender optimization is NP-hard, we use a two-regime solver for client-level defense selection (not aggregation redesign):

*Exact feasible-set search* for small instances ($K \leq 15$) evaluates all budget-feasible defense sets and selects the one with minimum estimated residual damage.

*Candidate-based search* for larger instances ($K > 15$) first generates feasible defense candidates (a ratio-greedy seed plus randomized feasible subsets), then evaluates each candidate under the same Stackelberg objective via Monte Carlo estimation of $\mathcal{H}(x) = \mathbb{E}_o[D(x, o)]$, and returns the lowest-damage set. The estimator uses a sequential sample-size rule targeting $\varepsilon = 0.02$ absolute error at $\beta = 0.05$ failure probability (using $\beta$ to avoid collision with defense strength $\delta$), with sample count bounded in $[100, 1000]$.

The attacker oracle, used inside both regimes, solves the knapsack problem in Lemma 2 via exact dynamic programming when tractable, and otherwise falls back to greedy+best-single with a $\frac{1}{2}$-approximation guarantee (Algorithm 1).

---

**Algorithm 1** Attacker Oracle

---

**Require:** Expected values $\{v_i\}_{i=1}^K$, attack costs $\{c_{a,i}\}_{i=1}^K$, budget $B_A$
**Ensure:** Attack set $s_A^*$
 1: Keep profitable items $I^+ \leftarrow \{i : v_i > 0\}$
 2: Attempt exact integer-scaled DP on $(I^+, v_i, c_{a,i}, B_A)$
 3: **if** exact DP succeeds **then**
 4:   **return** exact optimal $s_A^*$
 5: **else**
 6:   Compute density-greedy set $s_A^{(g)}$
 7:   Compute best feasible singleton $s_A^{(1)}$
 8:   **return** $\arg\max \left\{ \sum_{i \in s_A^{(g)}} v_i, \ \sum_{i \in s_A^{(1)}} v_i \right\}$
 9: **end if**

---

**Complexity:** Exact DP is pseudo-polynomial in the scaled budget, while fallback greedy+best-single is $O(|I^+| \log |I^+|)$ and preserves the $\frac{1}{2}$ approximation guarantee.

For the defender, the implementation evaluates *expected residual damage* via Monte Carlo, where each sample calls the attacker oracle with a noisy observation realization.

*Exact Feasible-Set Search (Small Systems).* For systems with $K \leq 15$, all budget-feasible defense sets are enumerated and scored (Algorithm 2):

*Candidate Search (Larger Systems).* For $K > 15$, we avoid exhaustive enumeration: construct a greedy seed defense by sorting clients via $w_i/c_{d,i}$ and packing under $B_D$, generate additional randomized feasible candidates, score each with the same Monte Carlo damage estimator, and select the minimum. The estimator uses pilot samples and confidence-controlled sample sizing (bounded between implementation minima and maxima).

Algorithm 3 summarizes the complete FRL-CDPS training loop, showing how the defender solver, attacker oracle, and FRL updates interact within each federated round.

---

**Algorithm 2** Exact Defender Search

---

**Require:** Client set $\mathcal{C}$, costs $\{c_{d,i}\}$, budget $B_D$, MC estimator
**Ensure:** Defense set $s_D^*$
1: Enumerate $\mathcal{F}_D = \{s_D \subseteq \mathcal{C} : \sum_{i \in s_D} c_{d,i} \leq B_D\}$
2: $J^* \leftarrow +\infty, s_D^* \leftarrow \emptyset$
3: **for** each $s_D \in \mathcal{F}_D$ **do**
4:     Estimate $\widehat{J}(s_D) = \mathbb{E}[\text{damage} \mid s_D]$ by Monte Carlo
5:     **if** $\widehat{J}(s_D) < J^*$ **then**
6:         $J^* \leftarrow \widehat{J}(s_D); s_D^* \leftarrow s_D$
7:     **end if**
8: **end for**
9: **return** $s_D^*$

---

---

**Algorithm 3** FRL-CDPS: Stackelberg Defense for Federated Reinforcement Learning

---

**Require:** Clients $\mathcal{C} = \{1, \ldots, K\}$; game parameters $\{w_i, c_{d,i}, c_{a,i}\}$; budgets $B_D, B_A$; observation accuracy $q$; defense strength $\delta$; reshuffle frequency $T_{\text{reshuffle}}$; total rounds $T$
**Ensure:** Trained global policy $\pi_{\theta(T)}$
1: Initialize global policy $\pi_{\theta(0)}$
2: **Compute initial Stackelberg defense:** $s_D^* \leftarrow \text{DEFENDERSOLVER}(\{w_i, c_{d,i}, c_{a,i}\}, B_D, B_A, q, \delta)$
3: **for** $t = 1, 2, \ldots, T$ **do**
4:     **if** $t \bmod T_{\text{reshuffle}} = 0$ **then**
5:         Resample $\{w_i, c_{d,i}, c_{a,i}\}$; recompute $s_D^* \leftarrow \text{DEFENDERSOLVER}(\cdots)$
6:     **end if**
7:     *// Each client collects trajectories and computes gradients*
8:     **for** each client $i \in \mathcal{C}$ **in parallel do**
9:         Collect local trajectories $\mathcal{D}_i^{(t)}$ under $\pi_{\theta(t-1)}$; compute gradient $g_i^{(t)}$
10:     **end for**
11:     *// Attacker observes noisy defense signals and selects attack set*
12:     For each $i$: sample $o_i \sim P(\cdot \mid x_i)$ with $x_i = \mathbf{1}[i \in s_D^*]$
13:     Compute $\hat{p}_i(o_i) = 1 - \delta \cdot P(x_i{=}1 \mid o_i)$, then $\hat{v}_i(o_i) = w_i \hat{p}_i(o_i)$ for all $i$
14:     $s_A^* \leftarrow \text{ATTACKERORACLE}(\{\hat{v}_i\}, \{c_{a,i}\}, B_A)$                (Lemma 2)
15:     *// Adversarial clients corrupt gradients; defense reduces success probability*
16:     **for** each $i \in s_A^*$ **do**
17:         With probability $1 - \delta \cdot x_i$: inject gradient corruption on client $i$
18:     **end for**
19:     *// Server aggregates and updates global policy*
20:     $\theta^{(t)} \leftarrow \theta^{(t-1)} + \gamma \cdot \frac{1}{K} \sum_{i=1}^{K} g_i^{(t)}$          (FedAvg, $\gamma = $ learning rate)
21: **end for**
22: **return** $\pi_{\theta(T)}$

---

**Implementation notes.** Attack and defense costs are clipped away from zero to avoid degeneracies in density-based routines. The exact knapsack solver rescales real-valued costs/budgets to integers and uses memory-aware DP variants. All defender scores are Monte Carlo estimates under noisy observations computed with fixed seeds for reproducibility.

In summary, the framework establishes equilibrium existence (Lemma 1), proves that committing as leader weakly dominates any simultaneous-move Nash strategy (Claim 1), reduces the attacker's best response to a 0/1 knapsack (Lemma 2), proves NP-hardness of the defender's bilevel problem (Claim 3), and provides a $\frac{1}{2}$-approximation guarantee for the attacker oracle (Claim 2).

**Applicability of FRL-CDPS.** FRL-CDPS is directly applicable to FRL settings in which the server can commit to a client-level protection or validation set before receiving local updates, and where each client

Table 3: Complexity summary. Here $B'_A$ is the integer-scaled attack budget, $N_{MC}$ is Monte Carlo sample count, and $M$ is the number of defender candidates.

| Algorithm | Time Complexity | Space | Approximation |
|---|---|---|---|
| Attacker oracle (exact DP) | $O(|I^+| \cdot B'_A)$ | $O(|I^+| \cdot B'_A)$ | Optimal |
| Attacker oracle (fallback) | $O(|I^+| \log |I^+|)$ | $O(|I^+|)$ | $\frac{1}{2}$-approx. |
| Defender exact search | $O(|\mathcal{F}_D| \cdot N_{MC} \cdot T_{oracle})$ | $O(1)$ extra | Exact over $\mathcal{F}_D$ |
| Defender candidate search | $O(MK + M \cdot N_{MC} \cdot T_{oracle})$ | $O(MK)$ | Heuristic |

can be assigned scalar estimates of strategic importance and protection/attack cost. It can therefore be plugged into synchronous policy-gradient FRL pipelines using FedAvg or robust aggregation rules, because the Stackelberg layer acts before aggregation and does not require changing the optimizer or aggregation rule. It is not directly plug-and-play in settings where the server has no client-level intervention mechanism, clients are anonymous or fully peer-to-peer, the attacker controls the server, or the system is asynchronous without an explicit model of update arrival and staleness. In those cases, the game must be extended to include participation uncertainty, timing effects, or identity uncertainty.

## 5 Results

### 5.1 Experimental Setup

**Environments.** We evaluate FRL-CDPS on three standard benchmarks: CartPole-v1 (discrete control) and the continuous-control MuJoCo tasks HalfCheetah-v2 and Walker2d-v5 from OpenAI Gym (Brockman et al., 2016). $K = 30$ parallel workers independently interact with separate environment instances, each collecting trajectory data under the shared global policy $\pi_\theta$. Workers compute local policy gradients and transmit them to the central server for FedAvg aggregation. CartPole-v1 uses REINFORCE (Williams, 1992), while HalfCheetah-v2 and Walker2d-v5 use a PPO-style clipped surrogate objective (Schulman et al., 2017) with clip parameter $\epsilon = 0.2$. Training runs for $T = 5000$ rounds. CartPole-v1 results are averaged over 10 seeds, HalfCheetah-v2 results over 10 seeds, and Walker2d-v5 results over 3 seeds. The continuous-control results should be interpreted with the usual variance caveats of policy-gradient training. All experiments use synchronous, full-participation FRL to isolate the client-level defense-allocation problem under policy-gradient updates. The empirical scope is therefore intentionally narrower than the full space of federated agent systems: broader agent-based, multi-agent, and asynchronous FRL applications are important directions, but are left outside the present benchmark suite and discussed further in Section 6.

**Client heterogeneity.** Clients are assigned damage weights $w_i$, defense costs $c_{d,i}$, and attack costs $c_{a,i}$ reflecting realistic strategic heterogeneity: some clients are high-value targets with large $w_i$ and high defense costs, while others are cheaper to protect but less impactful. Default budgets are set as $B_D = 0.3 \times \sum_j c_{d,j}$ and $B_A = 0.3 \times \sum_j c_{a,j}$. Default observation accuracy is $q = 0.8$.

**Attack styles.** We instantiate three attack types: *gradient noise* (Bhagoji et al., 2019) (uniform noise perturbation to transmitted gradients), *action flip* (Huang et al., 2017; Lin et al., 2017) (adversarial action substitution during trajectory collection), and *reward poisoning* (Ma et al., 2019) (manipulated reward signals). Unless otherwise noted, experiments use gradient noise at intensity 0.5.

**Adversarial regimes.** We evaluate all methods under the four regimes from Definition 1: (i) single-client static, (ii) single-client reshuffling, (iii) multi-client static, (iv) multi-client reshuffling. In reshuffling regimes, game parameters are redrawn every $T_{reshuffle} = 100$ rounds, forcing learning-based defenses to continuously re-estimate client values while FRL-CDPS recomputes its Stackelberg allocation from the updated parameters.

**Metric.** We report mean cumulative episode reward as the primary metric, directly tracking $J(\pi_\theta)$. Tables additionally report ±std across seeds. Higher is better, and the unattacked clean baseline sets the performance ceiling.

### 5.2 Baselines

We compare FRL-CDPS against five baselines spanning both defense layers.

**Client-selection baselines** (all share the same budget $B_D$, cost structure, and defense mechanism as FRL-CDPS, so comparisons isolate the effect of the selection strategy):

- **Random:** Selects a budget-feasible protection set uniformly at random each round. Serves as a non-adaptive but unbiased baseline.

- **UCB:** Adapts the UCB-based client-selection schemes of Khajehali et al. (2025); Waref et al. (2025) to the defense setting. Each client $i$ maintains an empirical attack-frequency estimate $\hat{p}_i^{(t)}$ from noisy observations over $n_i^{(t)}$ rounds. The UCB score is:

$$s_i^{(t)} = w_i \left( \hat{p}_i^{(t)} + c\sqrt{\frac{\ln t}{n_i^{(t)}}} \right), \tag{22}$$

where $w_i$ is client $i$'s damage weight and $c > 0$ is the exploration constant. Each round, clients are ranked by $s_i^{(t)}$ and a budget-feasible subset is selected greedily. Counts reset on parameter reshuffle.

- **Thompson Sampling:** Adapts standard Beta-Bernoulli Thompson Sampling (Deressa & Hasan, 2024) to client defense selection. Each client $i$ maintains a Beta posterior over its attack probability, initialized as $\text{Beta}(1,1)$ and updated via:

$$\alpha_i^{(t+1)} = \alpha_i^{(t)} + \mathbf{1}[\text{attack observed on } i], \quad \beta_i^{(t+1)} = \beta_i^{(t)} + \mathbf{1}[\text{no attack observed on } i]. \tag{23}$$

Each round, a score $s_i = w_i \cdot \tilde{\theta}_i$ is computed from a sample $\tilde{\theta}_i \sim \text{Beta}(\alpha_i^{(t)}, \beta_i^{(t)})$, and a budget-feasible subset is selected greedily by descending score. Posteriors reset on reshuffle.

- **No Defense:** No client-level protection applied. Represents the lower bound.

**Server-side aggregation baselines** (composable with the above):

- **FLTG** (Wen et al., 2025): In the original paper, FLTG combines (i) a trusted or dynamically chosen reference update, (ii) ReLU-clipped cosine-similarity trust scores, (iii) norm alignment, and (iv) non-IID-aware weighting. Our FRL implementation instantiates the angle-based trust-weighting core using a Byzantine-robust proxy reference. Specifically, each client $i$ receives a trust score via ReLU-clipped cosine similarity to a reference gradient $\mathbf{g}_s$:

$$\text{TS}_i = \text{ReLU}\big(\cos(\mathbf{g}_i, \mathbf{g}_s)\big), \qquad \hat{\mathbf{g}}_i = \text{TS}_i \cdot \frac{\mathbf{g}_i}{\|\mathbf{g}_i\|} \cdot \|\mathbf{g}_s\|, \tag{24}$$

and aggregates $\mathbf{G} = \sum_i \hat{\mathbf{g}}_i / \sum_i \text{TS}_i$, downweighting updates that diverge from the trusted direction. In our FRL instantiation, where a separate labeled root dataset is unavailable, we use the coordinate-wise median of all received client gradients as a Byzantine-robust proxy for $\mathbf{g}_s$. Thus, the equation above matches our implementation of FLTG's angle-based component, but it does not claim to reproduce every mechanism in the original FLTG pipeline.

- **FedGreed** (Kritharakis et al., 2025): In the original paper, the server evaluates each client update on a trusted reference dataset, ranks clients by that loss, and greedily aggregates a low-loss subset. In our FRL instantiation, where no labeled root dataset is available, we replace this loss-based ranking with distance to a Byzantine-robust reference gradient. Let

$$\mathbf{g}_{\text{ref}} = \text{median}\big(\{\mathbf{g}_i\}_{i=1}^{K}\big), \qquad d_i = \|\mathbf{g}_i - \mathbf{g}_{\text{ref}}\|_2, \tag{25}$$

and let $\rho \in (0,1]$ denote the FedGreed selection fraction. We keep the $m = \max(1, \lfloor \rho K \rfloor)$ clients with smallest distances,

$$S_{\text{FG}} = \underset{S \subseteq \{1,\dots,K\},\, |S|=m}{\arg\min} \sum_{i \in S} d_i, \qquad \mathbf{G} = \frac{1}{m} \sum_{i \in S_{\text{FG}}} \mathbf{g}_i, \tag{26}$$

and aggregate only their gradients. This matches our implementation: the coordinate-wise median provides a Byzantine-robust reference, clients are ranked by Euclidean distance to that reference, and the closest fraction is averaged.

FRL-CDPS and the client-selection baselines operate at the pre-aggregation layer and are orthogonal to server-side rules. We evaluate both standalone and composed configurations.

**Implementation note.** Because our FRL setup does not provide a labeled trusted server dataset, all server-side baselines are instantiated in FRL-compatible form. For FLTG and FedGreed, this means replacing the original trusted-data scoring component with a Byzantine-robust reference-gradient proxy, while preserving the core trust-weighting or greedy trusted-subset logic used by each method.

**Computational resources.** All experiments were run on a single NVIDIA A100 80GB GPU. Each full training run ($T = 5000$ rounds, $K = 30$ workers) takes approximately 2–3 hours on CartPole-v1 and several hours on the continuous-control benchmarks HalfCheetah-v2 and Walker2d-v5. The Stackelberg solver (defender search + attacker oracle) operates on scalar game parameters rather than model parameters; per-round runtime relative to vanilla FedAvg is reported in Table 4. CPU preprocessing and game parameter resampling were handled on an AMD EPYC 7742 64-core processor with 512 GB RAM.

**Runtime comparison.** To quantify the computational cost of the proposed method relative to the baseline, Table 4 compares the clean vanilla-FedAvg baseline against FRL-CDPS on HalfCheetah-v2. We report the mean round time of the clean baseline and the corresponding FRL-CDPS runtime under a conservative setting in which the Stackelberg defense is recomputed at every round.

Table 4: Runtime comparison on HalfCheetah-v2. FRL-CDPS is evaluated under per-round Stackelberg recomputation (100 clients).

| Method | Mean Round Time |
|---|---|
| Vanilla FedAvg (clean baseline) | 43.67 s |
| FRL-CDPS | 78.51 s |

### 5.3 Adversarial Performance Analysis

FRL-CDPS achieves the highest final reward on the reported benchmarks in Table 5, outperforming the next-best heuristic (Thompson Sampling) by a margin that widens as training progresses (Figure 2). The CartPole-v1 ablation suite in Table 6 further confirms this advantage across all four attack regimes. The gap between FRL-CDPS and Thompson Sampling reflects the key advantage of commitment: FRL-CDPS precomputes which clients to protect given anticipated attacker behavior, while Thompson Sampling wastes rounds re-exploring client values after every reshuffle. UCB performs comparably to No Defense, which we hypothesize is because UCB over-prioritizes high-cost top-tier clients early in each era, leaving mid-tier clients undefended during the initial exploration phase, exactly the clients the attacker exploits before UCB's estimates converge. The continuous-control results, including HalfCheetah-v2 and Walker2d-v5, exhibit the usual variance associated with policy-gradient training under multiple seeds. However, the relative ordering across strategies is consistent and confirmed by the learning curves in Figure 2.

**Composition with server-side defenses.** The right panels of Figure 2 show learning curves when client-level placement strategies are composed with server-side aggregation defenses (FLTG and FedGreed). We evaluate all six pairwise compositions (each of FRL-CDPS, Thompson, UCB paired with each of FLTG, FedGreed) alongside standalone FLTG and FedGreed. Because FRL-CDPS operates at the client-selection layer rather than redesigning the aggregation rule itself, we do not interpret these comparisons as "Stackelberg versus aggregation" in a winner-take-all sense. Instead, the practical question is whether strategic client-level planning improves over alternative client-selection rules on its own, and whether that advantage is preserved when composed with server-side defenses. The results below should be read through that complementary-layer lens.

Across all three environments, FRL-CDPS is the strongest standalone client-placement method. When combined with server-side aggregation defenses, FRL-CDPS remains highly competitive and usually gives the strongest composed configuration within each aggregation family. On CartPole-v1 and HalfCheetah-v2, FRL-CDPS + FedGreed is the best composed method, and FRL-CDPS + FLTG remains above the corresponding UCB- and Thompson-based FLTG compositions. On Walker2d-v5, the composed results are closer, with FRL-CDPS + FedGreed remaining the best FedGreed-based composition and Thompson Sampling + FLTG slightly exceeding FRL-CDPS + FLTG. Overall, these results support the main point that Stackelberg client-level placement is complementary to server-side aggregation, while the best pairing can depend on the environment and aggregation rule.

Table 5: Final cumulative episode reward (mean $\pm$ std) under multi-client reshuffling ($T_{\mathrm{reshuffle}} = 100$) with gradient-noise attack at intensity 0.5.

| Strategy | CartPole-v1 | HalfCheetah-v2 | Walker2d-v5 |
|---|---|---|---|
| Clean Baseline (no attack) | $383.9 \pm 35.8$ | $547.7 \pm 169.7$ | $274.4 \pm 53.9$ |
| FRL-CDPS (Stackelberg) | $\mathbf{253.6 \pm 28.8}$ | $\mathbf{486.0 \pm 155.0}$ | $\mathbf{220.3 \pm 44.0}$ |
| Thompson Sampling | $212.8 \pm 38.9$ | $412.7 \pm 151.9$ | $175.1 \pm 33.2$ |
| Random | $207.0 \pm 38.1$ | $396.9 \pm 158.3$ | $143.7 \pm 27.9$ |
| UCB | $199.0 \pm 43.6$ | $372.3 \pm 160.2$ | $101.0 \pm 64.3$ |
| No Defense | $194.3 \pm 43.5$ | $342.0 \pm 159.7$ | $93.1 \pm 71.4$ |

## 5.4 Ablation Studies

To understand the sensitivity of the Stackelberg advantage, we conduct ablations varying:

- **Reshuffle frequency** $T_{\mathrm{reshuffle}} \in \{50, 100, 500\}$ (Figure 3a): at each reshuffle, damage weights $w_i$, defense costs $c_{d,i}$, and attack costs $c_{a,i}$ are jointly resampled, simulating realistic shifts in client characteristics (e.g., hardware changes, network topology updates). Shorter eras force bandit methods to relearn more often, widening the gap with Stackelberg planning. The static (no-reshuffle) regime is covered in Table 6.

- **Attack style** $\in$ {grad noise, action flip, reward poison} (Figure 3b): tests robustness to different corruption mechanisms.

- **Budget configuration** (Figure 4a): defender-favored ($B_D$=0.5, $B_A$=0.2), balanced ($B_D$=$B_A$=0.3), and attacker-favored ($B_D$=0.2, $B_A$=0.5).

- **Number of clients** $K \in \{20, 30, 50, 100\}$ (Figure 4b): tests scalability of FRL-CDPS's solver variants.

- **Attack intensity** $\in \{0.3, 0.5, 0.7\}$ (Figure 5a): robustness across mild and severe attack regimes.

- **Observation accuracy** $q \in \{0.6, 0.8, 1.0\}$ (Figure 5b): higher $q$ gives the attacker cleaner reconnaissance, paradoxically helping bandit defenses learn more accurate attack signals, while FRL-CDPS accounts for $q$ directly in its Bayesian posterior without relearning attack frequencies.

- **Attack regime** (Table 6): all four combinations of cardinality (single vs. multi-client) and persistence (static vs. reshuffling), isolating how much of the Stackelberg advantage comes from each axis.

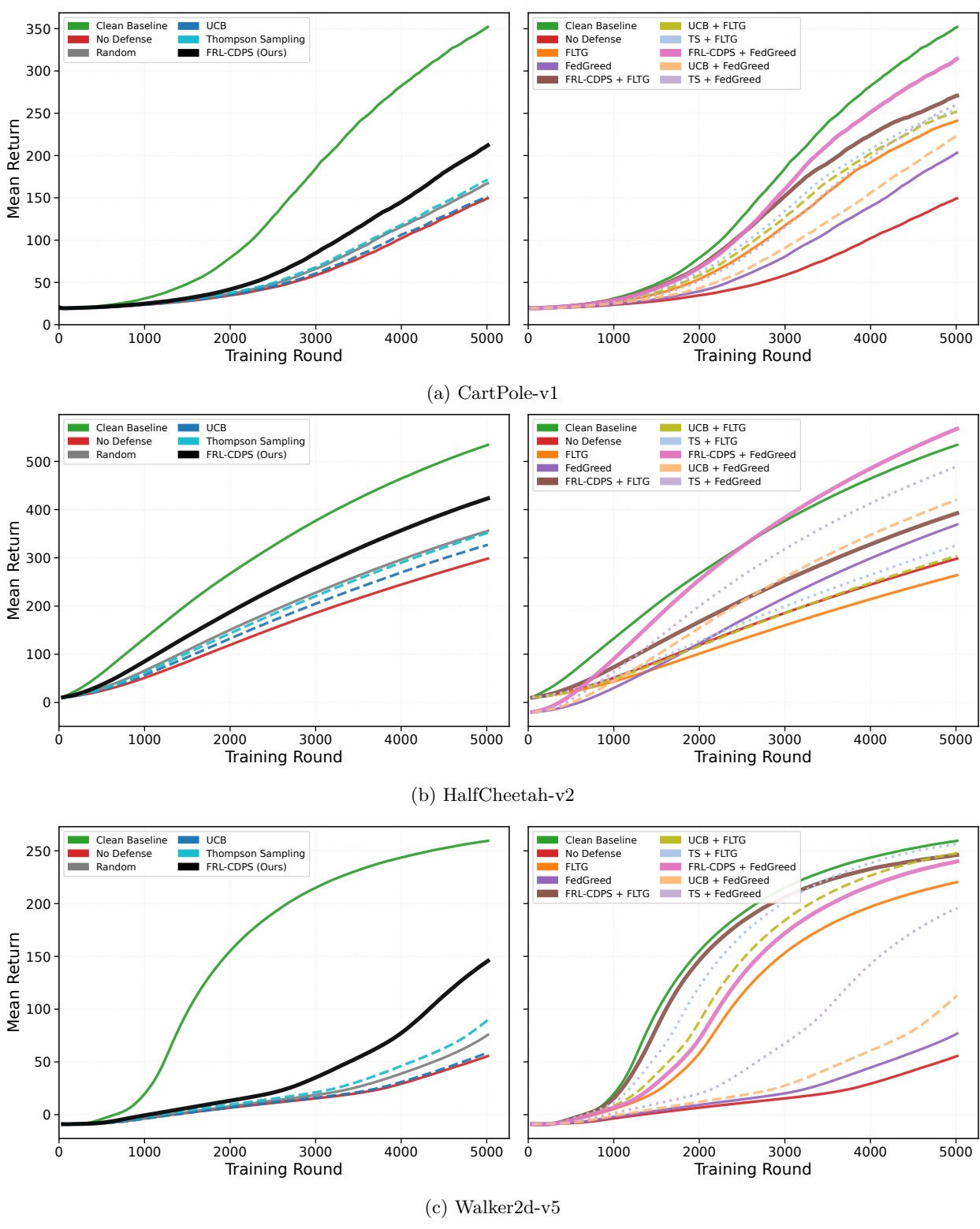

(a) CartPole-v1

(b) HalfCheetah-v2

(c) Walker2d-v5

Figure 2: Learning curves under multi-client reshuffling with gradient-noise attacks. *Top:* CartPole-v1. *Middle:* HalfCheetah-v2. *Bottom:* Walker2d-v5. In each benchmark, the left panel shows client placement strategies (standalone) and the right panel shows compositions with server-side aggregation defenses (FLTG and FedGreed).

**Reshuffle frequency & attack style.** Figure 3a shows that at low $T_{\text{reshuffle}}$, bandit methods incur systematic exploration penalties each era as they re-estimate client values from scratch after each parameter reshuffle, while FRL-CDPS recomputes its Stackelberg allocation directly from the updated $\{w_i, c_{d,i}, c_{a,i}\}$ without any exploration overhead. Figure 3b shows FRL-CDPS's advantage is consistent across gradient noise, action flip, and reward poisoning attacks: the relative ranking of all methods is preserved regardless of the corruption mechanism, confirming that the Stackelberg commitment benefit is attack-agnostic rather than specific to any single threat model.

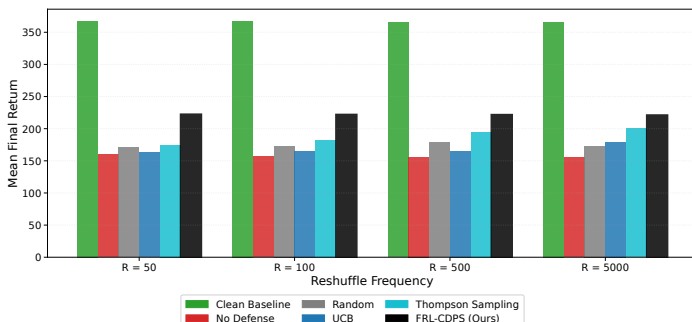 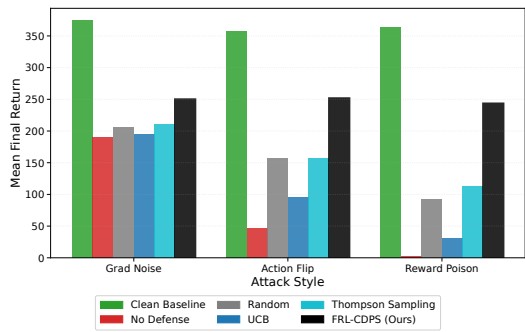

(a) Effect of reshuffle frequency. As $T_{\text{reshuffle}}$ decreases, bandit defenses incur larger exploration penalties while FRL-CDPS recomputes its Stackelberg allocation directly.

(b) Robustness across attack styles. FRL-CDPS consistently outperforms baselines regardless of the corruption mechanism.

Figure 3: Ablation: reshuffle frequency and attack style.

**Budget configuration & scalability.** Figure 4a shows FRL-CDPS degrades most gracefully under attacker-favored budgets ($B_A = 0.5$, $B_D = 0.2$). Under defender-favored budgets, all methods improve, but the relative ranking is preserved. Under attacker-favored conditions, heuristic methods drop sharply because they cannot anticipate which high-value clients the well-funded attacker will concentrate on, while FRL-CDPS explicitly accounts for the attacker's expanded reach when computing its protection set. Figure 4b shows that FRL-CDPS maintains a consistent advantage across $K \in \{20, 30, 50, 100\}$ clients. As $K$ grows, the solver switches from exact feasible-set enumeration (tractable for $K \leq 15$) to candidate-based Monte Carlo search, which trades optimality for scalability. Despite this approximation, FRL-CDPS continues to outperform heuristics because the Stackelberg commitment benefit persists even under approximate optimization.

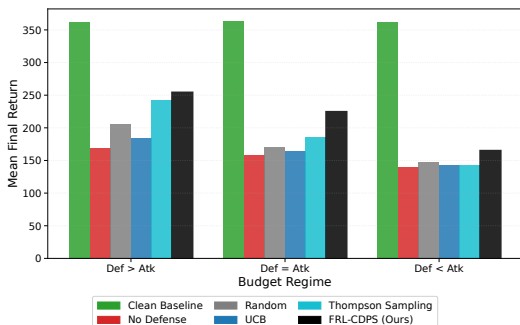 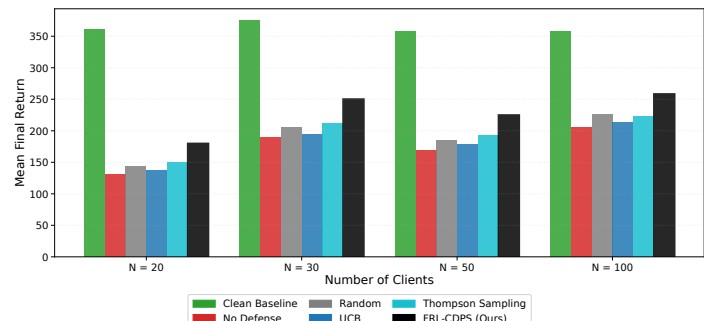

(a) Effect of budget asymmetry. FRL-CDPS maintains its advantage even when the attacker holds a larger budget.

(b) Scalability with number of clients $K$. FRL-CDPS's candidate-based solver maintains competitive performance as $K$ grows.

Figure 4: Ablation: budget configuration and scalability.

**Attack intensity & observation accuracy.** Figure 5a shows FRL-CDPS degrades most gracefully as attack magnitude $\eta$ increases from 0.3 to 0.7. At high intensity, uncovered clients suffer proportionally larger

gradient corruptions, so the value of correctly identifying and protecting high-damage-weight clients compounds, directly benefiting FRL-CDPS's principled prioritization over heuristic methods. Figure 5b shows a counterintuitive pattern: as attacker observation accuracy $q$ increases from 0.6 to 1.0, Thompson Sampling and UCB improve alongside FRL-CDPS. Higher $q$ gives the attacker cleaner signals about which clients are defended, making its attack patterns more predictable and consistent. This regularity in turn provides bandit defenses with cleaner feedback about which clients are being targeted, accelerating posterior convergence. FRL-CDPS accounts for $q$ directly in its Bayesian posterior (Equations 12–13) without relearning attack frequencies, maintaining a stable advantage across all $q$ values.

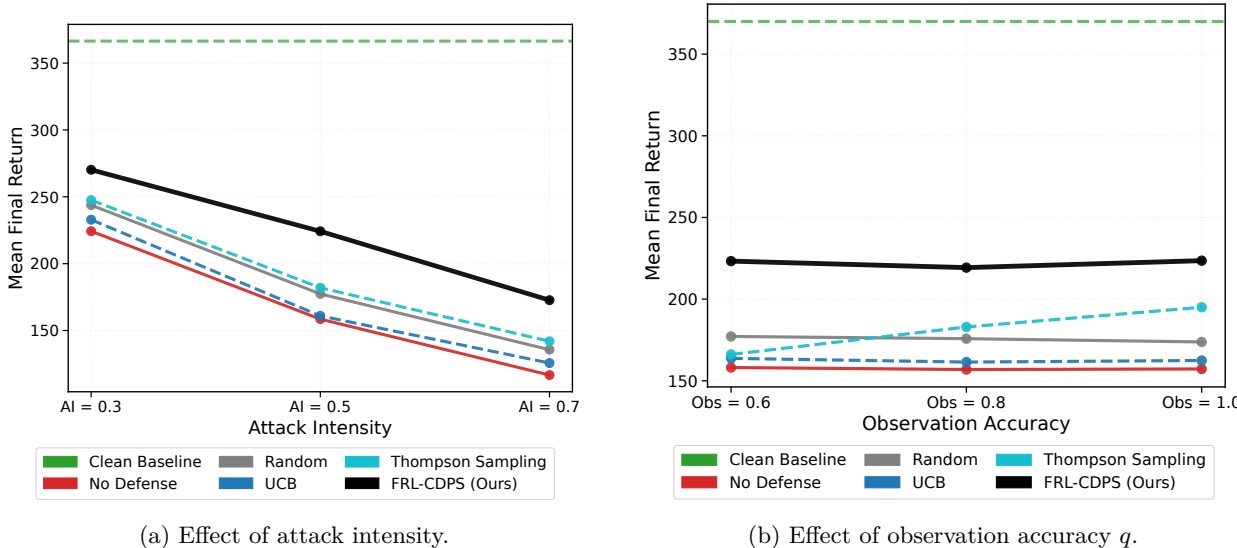

(a) Effect of attack intensity.       (b) Effect of observation accuracy $q$.

Figure 5: Ablation: attack intensity and observation accuracy.

**Single vs. multi-client attacker.** Table 6 compares all four regimes: cardinality (single vs. multi-client) × persistence (static vs. reshuffling). Under single-client attacks, all methods improve since only one client is corrupted per round, and even random defense has a reasonable probability of covering the active attacker. The static multi-client regime shows a moderate FRL-CDPS advantage: with fixed parameters, bandit methods eventually converge to good coverage, but FRL-CDPS begins from a game-aware placement immediately. The reshuffling multi-client regime (matching Table 5) shows the largest gap, as parameter resets force bandit methods to re-explore while FRL-CDPS recomputes from the updated game parameters. The static vs. reshuffling contrast within each cardinality directly quantifies how much of the Stackelberg advantage comes from non-stationarity robustness.

Table 6: Final reward (mean ± std, 10 seeds, CartPole-v1) across all four adversarial regimes. Static: game parameters fixed. Reshuffle: parameters resampled every $T_{\text{reshuffle}} = 100$ rounds.

| Strategy | Single Static | Single Reshuffle | Multi Static | Multi Reshuffle |
|---|---|---|---|---|
| FRL-CDPS | **303.4±11.2** | **296.6±13.9** | **260.8±11.6** | **253.6±28.8** |
| Thompson | 300.9±4.1 | 288.6±11.8 | 229.5±18.7 | 212.8±38.9 |
| UCB | 302.6±10.3 | 284.0±12.2 | 218.9±14.2 | 199.0±43.6 |
| Random | 284.5±10.5 | 282.3±14.1 | 212.7±18.6 | 207.0±38.1 |
| No Defense | 276.8±11.4 | 277.6±15.7 | 175.2±11.5 | 194.3±43.5 |

# 6 Limitations and Future Work

A limitation of this work is the reliance on externally supplied estimates of damage weights $w_i$ and defense/attack costs $c_{d,i}$ and $c_{a,i}$ in the Stackelberg solver. Although Section 4 discusses how these quantities can be estimated from deployment signals, imperfect estimates would affect the quality of the computed allocation, but not the form of the underlying optimization problem; learning or refining these quantities online is therefore an important next step. The current model also assumes that an explicit defense-budget rule is supplied for each round or reshuffle era. Although our reshuffling experiments allow client costs and damage weights to change over time, the solver still optimizes each era under a specified budget constraint rather than choosing a horizon-level budget schedule, trading off soft resource penalties, or adapting the budget endogenously to privacy, compute, or participation constraints. Extending FRL-CDPS to these dynamic-budget settings is therefore an important direction.

Our empirical study focuses on policy-gradient FRL instantiated with REINFORCE and PPO-style updates. Extending the full pipeline to federated actor-critic methods with explicit critic/value-function updates is nontrivial because the attacker can corrupt two coupled learning channels: the actor update and the critic update. Poisoning the critic can distort temporal-difference targets and advantage estimates, which can then indirectly corrupt otherwise benign actor updates. As a result, a single scalar damage weight $w_i$ may no longer capture the full impact of compromising client $i$; one may need separate actor- and critic-side damage values, or a joint value measuring how critic error propagates into policy degradation. The defender would also need validation tests for both policy-gradient consistency and value-function consistency. FRL-CDPS can still serve as the client-allocation layer if such per-client attack values are available, but deriving and validating those values for federated actor-critic training requires a separate threat model and empirical study.

Our threat model focuses on adversaries whose objective is to degrade training through active corruption of the training signal (gradient noise injection, action flip, reward poisoning). We model reconnaissance only as the noisy defense-status signal used by the attacker for target selection. Attacks whose primary objective is passive information gathering or active probing of clients and network structure, without necessarily perturbing training dynamics, correspond to a different adversarial objective and are left as an out-of-scope but well-motivated future direction.

Finally, this paper studies synchronous, full-participation FRL. Asynchronous FRL introduces challenges absent from our current formulation. First, the defender may need to choose whom to protect before knowing which clients will actually return updates, making client availability part of the defense state. Second, stale gradients can differ statistically from fresh gradients, so anomaly signals may confound benign staleness with adversarial corruption. Third, an adaptive attacker could exploit timing by targeting stragglers, delaying compromised updates, or attacking clients whose stale gradients receive high aggregation weight. Extending FRL-CDPS to this setting would require augmenting the Stackelberg game with arrival probabilities, staleness-dependent damage weights, and timing-aware attacker utilities. A natural direction is to replace $w_i$ with a time-dependent value $w_i(t, \Delta_i)$ that accounts for update delay $\Delta_i$, and to solve defense placement over the expected set of arriving clients.

# 7 Conclusion

We presented FRL-CDPS, a Stackelberg game-based framework for budgeted client-level defense placement in federated reinforcement learning. By modeling the defender as a leader who commits to a protection strategy and the attacker as a follower who best-responds under imperfect reconnaissance, FRL-CDPS converts the inherently reactive problem of client-level defense into a proactive, anticipatory optimization. The key theoretical contributions (reduction of the attacker's best response to a 0/1 knapsack, NP-hardness of the defender's bilevel problem, and a $\frac{1}{2}$-approximation guarantee) ground the practical solvers in rigorous foundations.

Empirically, FRL-CDPS consistently outperforms heuristic client-selection baselines (random, UCB, Thompson sampling) across the full CartPole-v1, HalfCheetah-v2, and Walker2d-v5 benchmark suite. The advantage is most pronounced under frequent parameter reshuffling, which models realistic non-stationarity in

federated systems: client availability, hardware profiles, network conditions, vulnerability levels, and strategic importance can change over time. In this setting, bandit methods must re-explore client values after each shift, while FRL-CDPS recomputes its Stackelberg allocation directly from the updated game parameters. When composed with server-side aggregation defenses, FRL-CDPS remains a strong and often best-performing client-placement method, while the best composed configuration can vary with the aggregation rule and environment. Standalone server-side defenses provide only marginal improvement over no defense, confirming that client-level placement is the dominant robustness lever and that FRL-CDPS occupies a complementary layer in the FRL defense stack.

## 8 LLM Usage Declaration

We acknowledge the use of large language models (LLMs) to assist in refining portions of the text in this manuscript. Their use was strictly limited to improving readability, grammar, spelling, and style. All ideas, interpretations, and conclusions presented in this work are solely the responsibility of the authors.

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
