# OpenReview forum: "Client-Level Defense Placement for Adversarially Robust Federated Reinforcement Learning"
_TMLR — Accepted by TMLR_

### Review · Reviewer_jr2r · 2026-04-03

**Summary Of Contributions:**

Federated reinforcement learning (FRL) is an important research area, and attacks on policy gradient methods pose a serious concern that needs to be addressed. The authors consider an adaptive attack setting and combine both client- and server-level defenses. They propose FRL-CDPS, which models client-level defense allocation as a budget-constrained Stackelberg game. Experiments are conducted to demonstrate the effectiveness of the proposed algorithm.

Strengths:

1. The formulation of FRL as a Stackelberg game is interesting and provides a useful perspective for viewing FRL as an interactive system.
2. The authors consider the worst-case attacker by modeling the attacker’s best response.

Weaknesses:

1. The paper focuses on policy gradient methods and suggests that the framework can be extended to PPO. It would be valuable to clarify whether the approach generalizes to Actor–Critic methods (e.g., Federated Actor-Critic [1]), where the Q-function is updated using trajectories. In particular, how would the method handle adversarial perturbations to the Q-function?
2. [Minor] The assumption of a bounded defense budget may be restrictive. While budget constraints are commonly used in differential privacy for normalization, in practice, clients may have flexible or effectively unbounded budgets. For example, in federated settings, a client may initially participate actively but later withdraw as privacy costs accumulate. In such cases, the notion of a fixed budget may not fully capture realistic scenarios.
3. Although CartPole-v1 and HalfCheetah-v2 are standard benchmarks, the authors may consider evaluating on more recent or diverse reinforcement learning tasks to further strengthen the empirical validation.

Reference:
[1] Yang, T., Cen, S., Wei, Y., Chen, Y., & Chi, Y. (2024). Federated natural policy gradient and actor critic methods for multi-task reinforcement learning. *Advances in Neural Information Processing Systems*, 37, 121304–121375.

**Audience:**

No

**Audience Explanation:**

Federated reinforcement learning is an important research area; however, focusing only on standard policy gradient methods and PPO may limit the scope of the work. In principle, the proposed method could extend to more general federated agent settings, but this would require additional empirical validation [2]. Exploring such extensions could broaden the impact and appeal of the paper.

While it is reasonable to focus on synchronous FRL, a more detailed discussion of asynchronous settings, along with potential directions for future work, would further strengthen the paper and attract a wider audience.

Reference:
[2] Chen, C., Zhu, K., Chen, Z., Zhou, Z., Diao, S., Lu, Y., Li, T., Li, M., & Song, D. *Federated Agent Reinforcement Learning*. In *LLM-based Multi-Agent Systems: Towards Responsible, Reliable, and Scalable Agentic Systems*.

**Broader Impact Concerns:**

No ethical concerns are identified. The AI usage declaration is included.

**Claims And Evidence:**

No

**Claims Explanation:**

Some points raised in the weaknesses remain unclear. In addition to the aforementioned concerns, I have the following questions:

1. In the threat model, how many clients are assumed to be adversarial? Is this based on a Byzantine attack assumption (i.e., a fraction of malicious clients), or do the authors assume that even a single attacker is sufficient?

2. Does the server have any capability to detect adversarial behavior? For example, can it identify poisoned gradients, rewards, or actions? If not, could the authors clarify why such anomalies cannot be detected by the server?

3. In the threat model, do the authors consider the possibility that a client may switch from a benign participant to an attacker? This could occur either intentionally or as a consequence of the system being compromised. For instance, if an attacker uploads malicious updates that corrupt the global policy, a previously legitimate client might begin to follow and propagate these incorrect policies, effectively behaving as an attacker. It would be helpful to clarify whether such dynamics are captured in the Stackelberg formulation, as this scenario does not appear to be explicitly discussed.

**Requested Changes:**

- Threat Model:

1. What assumptions are made about the clients (e.g., how many are adversarial)?
2. Do the authors consider the possibility that a client may transition from a benign participant to an attacker (e.g., due to system compromise)?
3. Conversely, can an attacker revert to a benign role, or are roles assumed to be fixed?
4. Does the server have the capability to distinguish between benign and adversarial clients?

- Applications:

1. Consider extending the evaluation to other FRL scenarios.
2. Include more recent reinforcement learning benchmarks.
3. Explore more modern RL applications, such as agent-based systems leveraging policy gradient methods.

- Discussion:

1. Provide discussion or future directions for asynchronous FRL settings.

- Writing and Presentation:

1. Please number all key formulations to improve readability and referencing. For example, in Section 3.1, variables such as (s), (a), and (r) appear to denote state, action, and reward, but clearer notation definitions would be helpful.
2. [Minor] The font size and line width in some figures are too small and should be improved for readability.
3. Consider moving Algorithm 3 earlier, as it is currently embedded within the evaluation section and disrupts the flow.
4. Introduce Section 4.4 and Table 3 earlier. At present, key notation definitions appear only after the methodology, which makes it harder to follow the paper.

---

> ### Author Response · Authors · 2026-04-24
> **Authors Response**
>
> We thank the reviewer for the constructive feedback. We respond point-by-point below.
>
> ---
>
> ## Threat Model
>
> > *In the threat model, how many clients are assumed to be adversarial? Is this based on a Byzantine attack assumption (i.e., a fraction of malicious clients), or do the authors assume that even a single attacker is sufficient?*
>
> **Answer:** We thank the reviewer for this clarifying question. The threat model does not fix a Byzantine fraction in advance. Instead, the number of adversarial clients is determined endogenously by the attacker's budget $B_A$ and per-client attack costs $c_{a,i}$. At each round, the attacker solves a budget-constrained best-response problem and selects the subset that maximizes expected damage. This gives both the single-client regime ($|s_A|=1$) and the multi-client regime ($|s_A|>1$, bounded by $B_A$), and we evaluate FRL-CDPS in both regimes.
>
> **Manuscript modifications.**
> - **Section 3.2 (Threat Model) and Definition 1:** clarified that the number of adversarial clients is budget-determined rather than fixed by a Byzantine fraction.
>
> ---
>
> > *Does the server have any capability to detect adversarial behavior? For example, can it identify poisoned gradients, rewards, or actions? If not, could the authors clarify why such anomalies cannot be detected by the server?*
>
> **Answer:** We thank the reviewer for raising this point. The server does not observe ground-truth benign/adversarial labels, raw local trajectories, rewards, or action sequences. It only receives the client-side updates sent for aggregation. As a result, poisoned rewards and action manipulations are not directly visible to the server, and poisoned gradients can only be screened through imperfect validation or robust aggregation rather than oracle detection. In our formulation, this limited detection capability is modeled probabilistically through the defense strength $\\delta \\in [0,1]$: defending a client activates a validation mechanism, such as gradient anomaly detection, that causes the attack to fail with probability $\\delta$. The server must therefore allocate its limited validation budget $B_D$ to the clients where it matters most. This is also why our comparisons include server-side aggregation defenses (FLTG, FedGreed) as composable baselines, rather than assuming that the server can reliably identify all adversarial updates.
>
> **Manuscript modifications.**
> - **Section 3.2 (Threat Model):** clarified that the server does not observe ground-truth adversarial labels and that detection is imperfect rather than oracle-based.
>
> ---
>
> > *Do the authors consider the possibility that a client may switch from a benign participant to an attacker, or vice versa?*
>
> **Answer:** We thank the reviewer for this question. Role switching in both directions is supported by the threat model. First, the attacker recomputes its budget-constrained best response every round, so the set of compromised clients can change across rounds: a previously benign client can become compromised later if selected, and a previously attacked client reverts to benign behavior in rounds where it is not selected. Second, the reshuffling regime resamples $\\{w_i, c_{d,i}, c_{a,i}\\}$ every $T_{\\mathrm{reshuffle}}$ rounds, which changes the attacker's optimal target set from era to era. A client that is a cost-effective target in one era may become unattractive in the next, and vice versa.
>
> Regarding the follow-on scenario of "role contagion", where a benign client becomes *intrinsically* malicious solely because the shared global policy has been degraded, this is not modeled in our formulation. Such clients remain benign in our threat model. Degraded policies may cause them to generate lower-quality trajectories, but this is a training-quality effect rather than a change in adversarial role.
>
> **Manuscript modifications.**
> - **Section 3.2 (Threat Model):** added explicit clarification that compromise status is not permanent across rounds, with both mechanisms (per-round best response and parameter reshuffling) spelled out; explicitly scoped out role contagion.

---

> > ### Author Response · Authors · 2026-04-24
> > **Authors Response**
> >
> > ## Extension to Actor–Critic Methods
> >
> > > *It would be valuable to clarify whether the approach generalizes to Actor–Critic methods (e.g., Federated Actor-Critic [Yang et al., 2024]), where the Q-function is updated using trajectories. In particular, how would the method handle adversarial perturbations to the Q-function?*
> >
> > **Answer:** We thank the reviewer for this suggestion and the reference to Yang et al. (2024). The Stackelberg client-allocation layer is not intrinsically tied to policy-gradient optimizers, since it operates on scalar game parameters (damage weights, costs, defense strength) rather than on a specific optimizer state. Extending the full pipeline to federated actor-critic methods would require modeling adversarial perturbations to both the actor and the critic. Corrupted TD targets or value/Q-function updates enter the training signal through a different channel, and client damage weights would need to be redefined in terms of joint actor-critic degradation rather than policy-gradient degradation alone.
> >
> > **Manuscript modifications.**
> > - **Section 6 (Limitations and Future Work):** added a paragraph on extension to federated actor-critic methods, outlining the required modeling changes on the actor and critic channels.
> >
> > ---
> >
> > ## Bounded Defense Budget
> >
> > > *The assumption of a bounded defense budget may be restrictive. In federated settings, a client may initially participate actively but later withdraw as privacy costs accumulate. In such cases, the notion of a fixed budget may not fully capture realistic scenarios.*
> >
> > **Answer:** We thank the reviewer for raising this scenario. Our formulation uses a bounded per-round (or per-era) defense budget because validating, auditing, or cryptographically securing every client every round incurs non-trivial computational, communication, and monetary costs in realistic federated deployments. In the experimental setting, the default budget is set as $B_D = 0.3 \\times \\sum_j c_{d,j}$, and under the reshuffling regime the costs $\\{w_i, c_{d,i}, c_{a,i}\\}$ are jointly resampled every $T_{\\mathrm{reshuffle}}$ rounds. Because $B_D$ is proportional to $\\sum_j c_{d,j}$, the effective per-era budget changes as the costs are resampled, partially capturing settings where protection costs evolve with participation, compute, or privacy conditions. In deployments with effectively unconstrained resources, $B_D$ can be set high enough to cover all clients, recovering full protection as a special case. Fully continuous, horizon-level soft-budget formulations, such as the privacy-budget accumulation scenario the reviewer describes, are a useful extension beyond the per-era mechanism used here.
> >
> > **Manuscript modifications.**
> > - **Section 6 (Limitations and Future Work):** added a sentence acknowledging fully continuous, horizon-level, or soft budget constraints as a future direction, motivated by evolving privacy, compute, and participation costs in real deployments.
> >
> > ---
> >
> > ## Benchmarks
> >
> > > *Although CartPole-v1 and HalfCheetah-v2 are standard benchmarks, the authors may consider evaluating on more recent or diverse reinforcement learning tasks to further strengthen the empirical validation. Also consider extending the evaluation to other FRL scenarios, including more modern RL applications such as agent-based systems.*
> >
> > **Answer:** We agree that a broader benchmark suite strengthens the empirical case. We have added Walker2d-v5 as a third benchmark environment, with main learning curves and final-reward comparisons across all three environments. Regarding more modern agent-based FRL settings such as those in Chen et al., our framework targets synchronous policy-gradient FRL, while the referenced settings involve substantially different architectures and communication structures, including LLM-based multi-agent systems and heterogeneous tool use. Extending the Stackelberg client-allocation layer to that regime is a well-motivated but distinct research direction.
> >
> > **Manuscript modifications.**
> > - **Abstract and Section 5.1 (Experimental Setup):** added Walker2d-v5 as a third benchmark environment.
> > - **Main results table and main learning-curve figure:** added Walker2d-v5 results.

---

> ### Author Response · Authors · 2026-04-24
> **Authors Response**
>
> ## Asynchronous FRL
>
> > *While it is reasonable to focus on synchronous FRL, a more detailed discussion of asynchronous settings, along with potential directions for future work, would further strengthen the paper.*
>
> **Answer:** We agree this is a valuable discussion to include. Asynchronous FRL introduces challenges that are absent from the current synchronous formulation: stale gradients, delayed or missing client updates, defense decisions under partial participation, and timing-aware attackers that may exploit stragglers or delayed observations. Extending Stackelberg defense placement to this setting requires coupling the client-selection game with update staleness and arrival processes.
>
> **Manuscript modifications.**
> - **Section 6 (Limitations and Future Work):** added a dedicated paragraph on asynchronous FRL and the coupling of client-selection with staleness and arrival processes.
>
> ---
>
> ## Writing and Presentation
>
> > *Please number all key formulations; introduce Section 4.4 and Table 3 earlier; move Algorithm 3 earlier so it is not embedded in the evaluation section; the font size and line width in some figures are too small.*
>
> **Answer:** We thank the reviewer for these suggestions.
>
> - **Equation numbering.** All key equations in the revised manuscript are now numbered and referenced consistently throughout.
> - **Notation summary.** The notation summary table for FRL-CDPS has been moved to the beginning of Section 4 so that all symbols are defined before the methodology is presented.
> - **Main algorithm.** In the revised manuscript, the main FRL-CDPS training-loop algorithm appears at the end of Section 4 (Framework), before the Results section, so it is no longer embedded in the evaluation discussion.
> - **Figure readability.** In the revised manuscript, font sizes and line widths in the figures have been increased for readability.
>
> **Manuscript modifications.**
> - **Section 4 (Framework):** key equations numbered; notation summary table moved to the beginning of Section 4.
> - **Main FRL-CDPS training-loop algorithm:** placed at the end of Section 4, before the Results section.
> - **Figures:** font sizes and line widths have been increased for readability.
>
> ---
>
> We thank the reviewer again for the constructive comments. The revised manuscript addresses the main concerns raised in the review: it clarifies the threat model, adversarial client roles, and server detection assumptions; discusses actor-critic and asynchronous FRL extensions; adds Walker2d-v5 to broaden the benchmark suite; and improves the presentation through clearer notation, equation numbering, algorithm placement, and figure readability.

---

### Review · Reviewer_cXAB · 2026-04-13

**Summary Of Contributions:**

This paper proposes FRL-CDPS to model the client-level defense allocation in federated reinforcement learning (FRL) as a Stackelberg game between the defender and the attacker. This work develops some theoretical foundations of this formulation and provides some solution methods to obtain optimal defense strategies under budget constraints. This work also considers client-level layer, and finally conducts some exxperiments with ablations studies to showcase the efficiency of the proposed methods.

**Audience:**

Yes

**Audience Explanation:**

This work studies reinforcement learning which is highly relevant to the areas of TMLR

**Claims And Evidence:**

Yes

**Claims Explanation:**

I think most of the presentation is clear in this work. Here are my questions and concerns:

1. The problem setting in Section 3.1 is a bit hard to understand with the current version. For example, $t$ never shows up, and the trajectory is defined sometimes as $\tau$ and sometimes as $D_i$. Is each client's trajectory identically distributed? A more detailed problem setting presentation will make this work stronger.

2. For the treat model, do you only consider the three type of corruptions? In practice the real attacked may corrupt the result in more different ways.

3. Is the defense and attack cost always the same across the time horizon for a single client?

4. For the experimental results, it would be better if authors could report the running time of the proposed method compared with baseline.

5. It seems that the theoretical contributions of this work are based on the prior work, and the proof is brief. Can authors use a paragraph to summarize the theoretical novelty of this work over existing ones?

6. For the server aggregation strategies, it seems that the Stackelberg formulation does not yield advantages over the baselines. And as the authors mention in the limitation, it might not be practical to assume some hyperparameters such as cost parameters are known to the user in reality. These limits the practical advantage of this work.

**Requested Changes:**

Please see the above section for my questions and concerns.

---

> ### Author Response · Authors · 2026-04-24
> **Authors Response**
>
> We thank the reviewer for the constructive feedback. We address each concern below.
>
> ---
>
> > *The problem setting in Section 3.1 is a bit hard to understand. The trajectory notation is inconsistent, and it is unclear whether each client's trajectory is identically distributed.*
>
> **Answer:** We thank the reviewer for flagging the inconsistent notation. We use the two trajectory notations for different purposes. A trajectory collected by client $i$ at round $t$ is denoted by $\\tau_{i,n}^{(t)}$, where $n$ indexes trajectories in client $i$'s local batch and $\\ell$ indexes within-trajectory time steps. The generic $\\tau$ is used only as a dummy trajectory variable when defining the expected-return objective. In the clean baseline, client rollouts are identically distributed conditional on the current global policy because all clients interact with independent copies of the same task; heterogeneity enters through the strategic parameters and through which clients are attacked or defended.
>
> **Manuscript modifications.**
> - **Section 3.1 (Federated Reinforcement Learning Setup):** rewritten with consistent trajectory notation, explicit client/sample/round indexing, and a clear statement that local rollouts are i.i.d. conditional on the global policy.
>
> ---
>
> > *For the threat model, do you only consider the three types of corruption? In practice, the real attacker may corrupt the result in more different ways.*
>
> **Answer:** We thank the reviewer for raising this point. The three attack types (gradient noise injection, action flip, reward poisoning) are intended to cover the principal client-side corruption surfaces in FRL: post-computation gradient corruption, pre-computation trajectory corruption, and reward manipulation. They are not intended to be an exhaustive taxonomy of all possible FRL attacks. The Stackelberg client-allocation layer can accommodate other client-side corruption models as long as they induce a per-client attack value and a realized residual-damage effect under the defense.
>
> **Manuscript modifications.**
> - **Section 3.2 (Threat Model):** added a sentence clarifying that the three attack types are representative instantiations and that the Stackelberg allocation layer accommodates other corruption models.
>
> ---
>
> > *Is the defense and attack cost always the same across the time horizon for a single client?*
>
> **Answer:** We thank the reviewer for this question. The costs are fixed over time only in the static regime. The paper studies two regimes along a persistence axis: in the static regime, the tuples $(w_i, c_{d,i}, c_{a,i})$ are fixed for all rounds; in the reshuffling regime, they are piecewise constant within each era and jointly resampled every $T_{\\mathrm{reshuffle}}$ rounds. This second regime models non-stationarity in federated deployments, such as changing client hardware, network conditions, or participation costs. Both regimes are evaluated in the experiments.
>
> **Manuscript modifications.**
> - **Section 3.2 (Threat Model) and Definition 1:** clarified the static and reshuffling regimes, including that costs are fixed in the static regime and piecewise constant within each reshuffling era.
>
> ---
>
> > *It would be better if authors could report the running time of the proposed method compared with the baseline.*
>
> **Answer:** We thank the reviewer for this suggestion. We now report mean per-round runtime for vanilla FedAvg (43.67 s) and FRL-CDPS (78.51 s) on HalfCheetah-v2 with 100 clients. This is a conservative measurement because the Stackelberg defense is recomputed at every round. In practice, when parameters are reshuffled every $T_{\\mathrm{reshuffle}}$ rounds, recomputation is only needed at era boundaries, so the amortized per-round overhead is lower. The solver operates on scalar game parameters rather than model parameters, which keeps its absolute cost independent of policy size.
>
> **Manuscript modifications.**
> - **Section 5.2 (Baselines):** added a runtime comparison table reporting mean per-round time for vanilla FedAvg and FRL-CDPS on HalfCheetah-v2.

---

> ### Author Response · Authors · 2026-04-24
> **Authors Response**
>
> > *It seems that the theoretical contributions of this work are based on prior work, and the proof is brief. Can authors use a paragraph to summarize the theoretical novelty of this work over existing ones?*
>
> **Answer:** We clarify which pieces are borrowed from prior work and which are new.
>
> The main theoretical novelty is the Stackelberg formulation of client-level defense placement in FRL. Classical Stackelberg security games provide the general leader-follower idea, and classical knapsack algorithms provide a way to solve a budgeted follower subproblem. FRL-CDPS uses these tools for a different object: the defender commits to a budgeted set of protected FRL clients before an adaptive attacker, observing only noisy reconnaissance signals, chooses which clients to compromise. Prior FL/FRL defenses mainly study server-side aggregation, detection/filtering, incentive or pricing mechanisms, or reactive client-selection heuristics. They do not formulate client-level protection placement in FRL as an anticipatory Stackelberg commitment problem under partial observability and probabilistic defense effectiveness.
>
> The supporting theoretical results make this Stackelberg formulation operational. The attacker best response reduces to a knapsack oracle only because the follower must choose a budget-feasible attack set after receiving a noisy defense-status signal. The defender's bilevel Stackelberg problem remains the central object: we prove its NP-hardness, provide exact and candidate-based solvers, and connect the game utilities to FRL policy degradation through client-specific damage weights. Thus, the knapsack component is a solver primitive for the attacker's best response, while the contribution is the Stackelberg defense-placement framework and its FRL-specific theoretical instantiation.
>
> To make this framing clear in the paper itself, the revised manuscript now includes a short paragraph in the theory section emphasizing Stackelberg client-level placement as the main theoretical contribution and positioning knapsack only as the follower-oracle component.
>
> **Manuscript modifications.**
> - **Section 4 (end of Framework):** added a short paragraph emphasizing that the theoretical novelty is the Stackelberg formulation of client-level defense placement in FRL, with the attacker knapsack serving as a follower-oracle component.
>
> ---
>
> > *For the server aggregation strategies, it seems that the Stackelberg formulation does not yield advantages over the baselines. And it might not be practical to assume some hyperparameters such as cost parameters are known to the user in reality.*
>
> **Answer:** We thank the reviewer for raising both concerns. FRL-CDPS operates at the client-selection layer and is designed to complement, not replace, server-side aggregation methods. Figure 2 reports two sets of comparisons. First, among client-selection strategies (Random, UCB, Thompson Sampling, FRL-CDPS), FRL-CDPS is consistently the strongest across benchmarks, both standalone and when composed with server-side defenses (FLTG, FedGreed). Second, standalone server-side defenses provide only marginal improvement over no defense, indicating that client-level placement is the dominant robustness lever in this setting. The empirical comparison should therefore be read as showing that FRL-CDPS is the strongest client-level defense and composes cleanly with server-side methods, rather than as a same-layer replacement for aggregation defenses.
>
> Regarding cost parameters, we agree that assuming known defense and attack costs $\\{c_{d,i}, c_{a,i}\\}$ is an idealization. In practice, defense costs can be specified from measurable deployment quantities such as compute overhead, validation cost, bandwidth, latency, or privacy/security budget consumed by protecting a client. Attack costs are harder to know exactly, but can be estimated from risk models, historical compromise patterns, vulnerability assessments, or conservative threat-model assumptions. The method does not require these estimates to be perfect: inaccurate costs may change the selected protection set, but the Stackelberg optimization remains the same and can be rerun when updated estimates are available. We clarify this assumption and discuss online refinement of these cost estimates in Section 6 of the revised manuscript.
>
> **Manuscript modifications.**
> - **Section 5.3 (Adversarial Performance Analysis):** clarified that FRL-CDPS is evaluated as a client-level placement layer that is complementary to server-side aggregation defenses.
> - **Section 6 (Limitations and Future Work):** added a sentence acknowledging the assumption of known cost parameters and online estimation as an important future direction.
>
> ---
>
> We thank the reviewer again for the constructive comments. The revised manuscript clarifies the setup, threat model, runtime, theoretical novelty, and the role of FRL-CDPS relative to server-side aggregation.

---

### Review · Reviewer_xeas · 2026-04-20

**Summary Of Contributions:**

Summary of Contributions

This paper proposes FRL-CDPS, which is a client-level defense algorithm against adversarial federated reinforcement learning. Formulating the problem as a stackelberg game, this paper assumes a partial observability to structure a defense against adversarial attacks. This paper has theoretical contributions, as well as verification on Cartpole and HalfCheetah benchmarks.

Strengths of this paper are as follows;
Full observability is often not available to the attackers and it is an interesting approach to formulate an defense around this feature. The theoretical formulations are also promising and could be expanded further.

Weakness of this paper are as follows;

The theory seems sound, but there are some assumptions that may limit scalability (defender procedure bing a candidate-based monte-carlo, iid assumptions for the attacker, etc). Combined with the fact that the paper was verified with relatively simple gym environments only, it would be better to show that the assumptions and the framework of the theory holds in much larger, complex environments.

As an additional note to above, not all attacks are directly targeted towards the distributed systems. Sometimes, it could be more of a probing attack to gather information about other systems in the network without leaving any trace or hindering the operation of the distributed system. It would be interesting to see how the algorithm holds in terms of scalability in numbers and scalability in terms of attack complexity.

Second, while it is expected that the attacker will have partial observability, it is difficult to expect how much the adversary can indeed observe. Therefore, it would be nice to have a sensitivity analysis on how much of the states are visible to the attacker until the defenses are compromised.

Thirdly, it appears that the scope of the experiment is too small. 2 experiments on relatively simple environments with some benchmarks only using 3 seeds seem a too small of a number to prove the novelty of the algorithm.

**Additional Comments:**

N/A

**Audience:**

Yes

**Audience Explanation:**

Defense against adversarial attacks is a topic of interest for the community, especially with more and more distributed / federated systems entering the market (local LLM, medical, finance, etc...).  This paper approaches the topic from partial observability, which is quite interesting. Researchers from robust / adversarial / multiagent community may find this approach interesting as well.

**Claims And Evidence:**

No

**Claims Explanation:**

The idea and the theory seems promising, but the experiments seems to be too simple to support the novelty. I have listed the details as weakness of the paper in the 'Summary of Contributions' section.

**Requested Changes:**

My requested changes are those pointed out as the weakness of the paper

1) Sensitivity analysis on how much of the states are observable until the safety is compromised

2) Study on scalability of the algorithm

3) More benchmarks and more seeds on the experiment

---

> ### Author Response · Authors · 2026-04-24
> **Authors Response**
>
> We thank the reviewer for the constructive feedback and for noting the value of the partial-observability perspective and theoretical formulation. We respond to each concern in the order raised by the review and summarize the corresponding manuscript changes.
>
> ---
>
> ## 1. Scalability Assumptions and Larger Environments
>
> > *The theory seems sound, but there are some assumptions that may limit scalability (defender procedure being a candidate-based Monte Carlo, i.i.d. assumptions for the attacker, etc). Combined with the fact that the paper was verified with relatively simple gym environments only, it would be better to show that the assumptions and the framework of the theory holds in much larger, complex environments.*
>
> **Answer:** We thank the reviewer for raising this important question. The two assumptions mentioned play different roles in the framework.
>
> - The **candidate-based Monte Carlo procedure** is the scalable defender solver used once exact feasible-set enumeration becomes impractical. This is the algorithmic mechanism used for large systems because the defender's bilevel problem is NP-hard.
> - The **i.i.d. prior** is a tractable Bayesian approximation in the attacker's belief model under partial reconnaissance. It makes the follower problem well-defined and computationally analyzable under uncertainty, rather than claiming that all realistic attackers must have exactly i.i.d. beliefs.
>
> The paper addresses scalability empirically in two complementary ways.
>
> - **Scalability in the number of clients.** Section 5.4 already includes a client-count ablation over $K \\in \\{20, 30, 50, 100\\}$, covering the transition from exact feasible-set search to candidate-based Monte Carlo search. FRL-CDPS maintains a consistent advantage across all tested values of $K$, showing that the Stackelberg benefit persists in the scalable solver regime.
> - **Scalability across environments.** In the revised manuscript, we have added **Walker2d-v5** in addition to CartPole-v1 and HalfCheetah-v2, so the empirical study is no longer limited to the original two-environment suite. This broadens the validation to a third control benchmark with a different continuous-control profile.
>
> Taken together, these results show that the framework remains effective both as the client population grows and across a broader continuous-control benchmark setting.
>
> **Manuscript modifications.**
> - **Section 5.1 (Experimental Setup):** added Walker2d-v5 as a third benchmark environment.
> - **Main results table and main learning-curve figure:** added Walker2d-v5 results.
>
> ---
>
> ## 2. Probing Attacks and Attack Complexity
>
> > *As an additional note to above, not all attacks are directly targeted towards the distributed systems. Sometimes, it could be more of a probing attack to gather information about other systems in the network without leaving any trace or hindering the operation of the distributed system. It would be interesting to see how the algorithm holds in terms of scalability in numbers and scalability in terms of attack complexity.*
>
> **Answer:** We thank the reviewer for raising this point. FRL-CDPS focuses on adversaries whose objective is to degrade training through active corruption of the training signal. In this setting, the paper studies three **representative and widely used** client-side corruption mechanisms:
>
> - **gradient noise injection**, covering post-computation gradient corruption,
> - **action flip**, covering trajectory-level action corruption,
> - **reward poisoning**, covering reward manipulation.
>
> These attacks were chosen because they span distinct and well-established FRL attack surfaces rather than reflecting a single narrow corruption model. The attack-style ablation in Section 5.4 shows that FRL-CDPS maintains its advantage across all three mechanisms.
>
> Stealthy **probing / reconnaissance** attacks correspond to a different adversarial objective: they aim to gather information without directly perturbing training dynamics. Since the utilities in FRL-CDPS are defined in terms of residual training damage, such attacks fall outside the current Stackelberg utility design. The revised manuscript now states this scope boundary explicitly.
>
> **Manuscript modifications.**
> - **Section 3.2 (Threat Model):** clarified that the three attacks are representative instantiations rather than an exhaustive taxonomy.
> - **Section 6 (Limitations and Future Work):** added a sentence acknowledging stealthy reconnaissance / probing attacks as outside the current threat model.

---

> ### Author Response · Authors · 2026-04-24
> **Authors Response**
>
> ## 3. Sensitivity to Partial Observability
>
> > *While it is expected that the attacker will have partial observability, it is difficult to expect how much the adversary can indeed observe. Therefore, it would be nice to have a sensitivity analysis on how much of the states are visible to the attacker until the defenses are compromised.*
>
> **Answer:** We thank the reviewer for this suggestion. This sensitivity analysis is already included in the paper through the **observation-accuracy ablation** in Section 5.4, where $q \\in \\{0.6, 0.8, 1.0\\}$ controls how accurately the attacker observes whether a given client appears defended.
>
> Here, partial observability refers to the attacker's noisy reconnaissance of **defense status**, not to visibility of the underlying MDP states. The observation-accuracy ablation therefore directly measures how the defense behaves as the attacker's reconnaissance becomes less or more accurate.
>
> FRL-CDPS accounts for $q$ algebraically through its Bayesian posterior and maintains a stable advantage across all tested values of $q$, while the bandit baselines vary with $q$ as the attacker's observations become more predictable.
>
> **Manuscript modifications.**
> - **Section 5.4 (Ablation Studies):** reports the observation-accuracy sensitivity analysis over $q \\in \\{0.6, 0.8, 1.0\\}$ and explains the resulting behavior.
>
> ---
>
> ## 4. Benchmarks and Seeds
>
> > *It appears that the scope of the experiment is too small. 2 experiments on relatively simple environments with some benchmarks only using 3 seeds seem too small to prove the novelty of the algorithm.*
>
> **Answer:** We thank the reviewer for this suggestion. To strengthen the empirical scope of the paper, we have added **Walker2d-v5** in the revised manuscript, so the benchmark suite now includes **CartPole-v1, HalfCheetah-v2, and Walker2d-v5**. The revised manuscript reports the corresponding main learning curves and final-reward comparisons on all three environments. The seed counts reported in Section 5.1 are **10** for CartPole-v1, **10** for HalfCheetah-v2, and **3** for Walker2d-v5.
>
> **Manuscript modifications.**
> - **Abstract and Section 5.1 (Experimental Setup):** added Walker2d-v5 as a third benchmark environment and updated the reported seed counts.
> - **Main results table and main learning-curve figure:** added Walker2d-v5 results.
>
>
> ---
> We thank the reviewer again for the constructive comments. The revised manuscript addresses the key concerns raised in the review: it includes explicit sensitivity analysis for partial observability, empirical scalability across larger client populations, robustness across multiple representative attack mechanisms, and a strengthened benchmark suite through the addition of Walker2d-v5.

---

> > ### Comment · Reviewer_xeas · 2026-05-03
> > **Thank you for your response!**
> >
> > The reviewer thanks the authors for responding to the questions and making the clarifications!

---

### Author Response · Authors · 2026-04-27
**Summary of changes in the revised manuscript**

We sincerely thank the reviewers for their thoughtful and constructive feedback. The reviews primarily focused on clarifying the threat model, strengthening the empirical scope, explaining scalability and runtime, clarifying the role of FRL-CDPS relative to server-side aggregation defenses, improving notation and presentation, and expanding the limitations discussion. In our rebuttal, we addressed all major concerns and specify the corresponding manuscript revisions as follows:

- **Reviewer jr2r + Reviewer cXAB - Threat model, adversarial clients, and server detection.** We clarify that the number of attacked clients is not fixed by a pre-specified Byzantine fraction. Instead, the attacker selects a budget-feasible subset according to its attack budget and per-client attack costs, which naturally covers both single-client and multi-client attack regimes. We also clarify that the server does not observe ground-truth adversarial labels, raw rewards, actions, or trajectories, and that detection is imperfect rather than oracle-based. This is modeled through the defense-strength parameter, where defending a client activates a local validation/protection mechanism that probabilistically reduces attack success. We further clarify that compromise status is not permanent, since the attacker recomputes its best response across rounds. These clarifications are incorporated in Section 3.2 (Threat Model) and Definition 1.

- **Reviewer cXAB + Reviewer xeas - Attack types, probing attacks, and attack complexity.** We clarify that gradient noise injection, action flip, and reward poisoning are representative instantiations of active FRL corruption, not an exhaustive taxonomy of all possible attacks. We also explicitly distinguish these active corruption attacks from stealthy probing or reconnaissance attacks, whose objective is information gathering without directly perturbing training dynamics. Such probing attacks are now identified as an out-of-scope but well-motivated future direction. These changes are made in Section 3.2 and in the Limitations / Future Work section.

- **Reviewer xeas + Reviewer jr2r - Benchmark scope, scalability, and seed reporting.** We strengthen the empirical evaluation by adding Walker2d-v5 as a third benchmark environment alongside CartPole-v1 and HalfCheetah-v2. The revised manuscript reports updated main results and learning curves for all three environments, and clarifies the seed counts used in each case. We also emphasize that scalability is evaluated along two axes: scaling the number of clients up to 100 and testing across a broader benchmark suite. These updates are incorporated in the Abstract, Section 5.1 (Experimental Setup), the main results table, and the main learning-curve figure.

- **Reviewer xeas - Partial observability and observation-accuracy sensitivity.** We clarify that partial observability refers to the attacker's noisy reconnaissance of client defense status, not to visibility of the underlying MDP state. The revised manuscript points to the observation-accuracy ablation over different values of the reconnaissance parameter, showing how FRL-CDPS behaves as attacker observations become more or less accurate. This sensitivity analysis is discussed in Section 5.4 (Ablation Studies), with the corresponding observation-accuracy figure.

- **Reviewer cXAB - Runtime transparency and implementation overhead.** We add a runtime comparison between vanilla FedAvg and FRL-CDPS on HalfCheetah-v2. The reported FRL-CDPS runtime uses a conservative setting in which the Stackelberg placement is recomputed every round. We also clarify that the solver operates on scalar game parameters rather than model parameters, so the placement computation is separate from policy-network size. These additions appear in Section 5.2 (Baselines / Runtime comparison) and the new runtime table.

- **Reviewer cXAB + Reviewer jr2r - Relationship to server-side aggregation defenses.** We clarify that FRL-CDPS is a client-level defense-placement framework, not an aggregation rule such as Krum, FLTG, or FedGreed, and not a standalone poisoning filter. Its role is to decide which clients receive an available local protection mechanism under a limited budget. We revised the discussion of composed experiments to emphasize that client-level placement and server-side aggregation are complementary layers. The comparisons with FLTG and FedGreed therefore test whether Stackelberg placement improves over alternative placement rules and whether that advantage is preserved when paired with aggregation defenses. These clarifications appear in the Introduction, Section 5.3 (Adversarial Performance Analysis), and the Conclusion.

---

> ### Author Response · Authors · 2026-04-27
> **Summary of changes in the revised manuscript**
>
> - **Reviewer cXAB - Theoretical novelty over prior work.** We add a clearer explanation of the theoretical contribution. The novelty is the Stackelberg formulation of budgeted client-level defense placement in FRL under partial observability and probabilistic defense effectiveness. The knapsack reduction is presented as the follower-oracle component that makes the framework operational, while the defender's bilevel placement problem remains the central object. This clarification is added in the Framework section after the theoretical results and solver discussion.
>
> - **Reviewer cXAB + Reviewer jr2r - Notation, algorithms, and presentation.** We revise the FRL setup with consistent trajectory notation, distinguishing a generic trajectory from client-, round-, and batch-indexed trajectories. We add a notation table at the beginning of the framework section, number and reference key equations consistently, and move the main FRL-CDPS training-loop algorithm before the Results section. We also improve figure readability and clean up presentation issues identified by the reviewers. These changes affect Section 3.1, Section 4, the main algorithm, the figures, and related equation/table references.
>
> - **Reviewer jr2r + Reviewer cXAB - Cost assumptions, budgets, actor-critic methods, and asynchronous FRL.** We expand the Limitations / Future Work section to discuss the assumption that damage weights and defense/attack costs are known or estimated. We clarify that imperfect estimates may affect allocation quality but not the form of the Stackelberg optimization, and identify online cost refinement as an important next step. We also discuss dynamic or soft defense budgets, extension to federated actor-critic methods where both actor and critic channels may be attacked, and asynchronous FRL settings involving stale gradients, delayed updates, partial participation, and timing-aware attackers.
>
> Overall, these changes substantially improve the clarity, scope definition, and presentation quality of the manuscript. We believe the revised version now makes the threat model more precise, strengthens the empirical evidence regarding scalability and robustness, clarifies the role of FRL-CDPS as a strategic client-level placement layer complementary to server-side aggregation, and resolves the presentation issues identified by the reviewers. We sincerely thank the reviewers again for their constructive feedback and believe these revisions make the paper significantly stronger.

---

### Decision · Action_Editor_zHWc · 2026-06-03

**Recommendation:** Accept with minor revision

**Additional Comments:**

The paper proposes FRL-CDPS, a defense against attacks on policy-gradient methods in federated reinforcement learning (FRL), combining client- and server-level defenses by modeling client-level defense allocation as a budget-constrained Stackelberg game with an adaptive, worst-case attacker. The reviewers recognize the importance of the problem, and highlight the Stackelberg formulation as an interesting and useful perspective. During discussion, the authors improved the work by clarifying the problem formulation and threat model, adding runtime comparisons and the Walker2d-v5 benchmark, and improving the writing. While a reviewer still identifies several broader-scope limitations, I judge the remaining actionable issues to be addressable through minor revision, and therefore recommend **minor revision**.


**Minor revision items**

* Discuss how attack/defense cost parameters could be estimated in real deployments.
* Summarize the theoretical novelty more clearly relative to prior work.
* Clarify the applicability of FRL-CDPS to broader FRL settings, including which settings it can or cannot be directly plugged into.
* Discuss asynchronous FRL scenarios, such as client selection and staleness, at least at the level of challenges and future directions.
* Add brief preliminary results for the federated actor-critic limitation, or provide a fuller discussion of why this extension is nontrivial and how it could be addressed.

**Audience:**

Yes

**Audience Explanation:**

The topic is highly relevant to TMLR, and the paper should attract sufficient interest within the ML community, particularly the federated learning community.

**Claims And Evidence:**

Yes

**Claims Explanation:**

The original submission raised several concerns, but the authors' revisions and their engagement with reviewers sufficiently addressed the majority of the issues. The latest version meets the decision criteria: claims supported by accurate, convincing, and clear evidence.

---

> ### Author Response · Authors · 2026-06-05
> **Revision Summary**
>
> We thank the Action Editor for the recommendation and for clearly identifying the remaining minor revision items. We appreciate the constructive guidance and have revised the manuscript to address each point as follows.
>
> - **Attack/defense cost estimation.** In **Section 4 (Framework of FRL-CDPS), Notation Summary, p. 8**, we added a new paragraph titled **“Estimating deployment parameters”**. This paragraph explains how the damage weights $\(w_i\)$, defense costs $\(c_{d,i}\)$, and attack costs $\(c_{a,i}\)$ can be estimated from deployment signals such as data volume, validation-return contribution, update influence, secure execution overhead, communication latency, device trust level, network exposure, patch status, and historical compromise alerts.
>
> - **Theoretical novelty relative to prior work.** In **Section 2 (Related Work), p. 4**, we added a dedicated paragraph titled **“Theoretical novelty relative to prior work”**. This clarifies that FRL-CDPS does not propose a new aggregation rule or reactive client-selection heuristic, but instead formalizes client-level defense placement as a finite Stackelberg security game under budget constraints, noisy reconnaissance, and probabilistic defense effectiveness. We also summarize the resulting theoretical contributions, including equilibrium existence, the attacker’s knapsack reduction, NP-hardness of the defender problem, and the \(1/2\)-approximation guarantee for the attacker oracle.
>
> - **Applicability to broader FRL settings.** We added a new paragraph titled **“Applicability of FRL-CDPS”** in **Section 4 (Framework of FRL-CDPS), p. 14**. This paragraph clarifies where FRL-CDPS can be directly plugged in, namely synchronous policy-gradient FRL pipelines where the server can commit to a client-level protection or validation set before receiving local updates. It also identifies settings where the method is not directly plug-and-play, including systems with no client-level intervention mechanism, anonymous or fully peer-to-peer clients, attacker-controlled servers, and asynchronous settings without explicit arrival/staleness modeling.
>
> - **Asynchronous FRL scenarios.** In **Section 6 (Limitations and Future Work), p. 22**, we expanded the limitations discussion to cover asynchronous FRL. The revised text discusses client availability, stale gradients, timing-aware attacks, arrival probabilities, and staleness-dependent damage weights. It also identifies a natural extension using time-dependent values $\(w_i(t,\Delta_i)\)$ for defense placement over the expected set of arriving clients.
>
> - **Federated actor-critic limitation.** We expanded **Section 6 (Limitations and Future Work), p. 22**, to provide a fuller explanation of why extending FRL-CDPS to federated actor-critic methods is nontrivial. The revised text explains that an attacker can corrupt both actor and critic updates, that critic poisoning can distort temporal-difference targets and advantage estimates, and that a single scalar damage weight $\(w_i\)$ may no longer be sufficient. We also discuss possible paths forward, including separate actor- and critic-side damage values and validation tests for both policy-gradient and value-function consistency.
>
> We are grateful for the Action Editor’s careful assessment and constructive suggestions. We believe these revisions address the remaining concerns and make the scope, assumptions, and practical applicability of FRL-CDPS clearer.